# Estimating time-dependent vegetation biases in the SMAP soil moisture product

Simon Zwieback[1,2], Andreas Colliander[3], Michael H. Cosh[4], José Martínez-Fernández[5],
Heather McNairn[6], Patrick J. Starks[7], Marc Thibeault[8], and Aaron Berg[1]

[1]Department of Geography, University of Guelph, Guelph, Ontario, Canada
[2]Department of Environmental Engineering, ETH Zurich, Zurich, Switzerland
[3]NASA Jet Propulsion Laboratory, California Institute of Technology, Pasadena, California, USA
[4]USDA-ARS Hydrology and Remote Sensing Laboratory, Beltsville, Maryland, USA
[5]Instituto Hispano Luso de Investigaciones Agrarias, Universidad de Salamanca, Salamanca, Spain
[6]Science and Technology Branch, Agriculture and Agri-Food Canada, Ottawa, Ontario, Canada
[7]USDA-ARS Grazinglands Research Laboratory, El Reno, Oklahoma, USA
[8]Comisión Nacional de Actividades Espaciales, Buenos Aires, Argentina

*Correspondence to:* Simon Zwieback (zwieback@uoguelph.ca)

**Abstract.** Remotely sensed soil moisture products are influenced by vegetation and how it is accounted for in the retrieval, which is a potential source of time-variable biases. To estimate such complex, time-variable error structures from noisy data, we introduce a Bayesian extension to triple collocation in which the systematic errors and noise terms are not constant but vary with explanatory variables. We apply the technique to the SMAP soil moisture product over croplands, hypothesizing that errors in the vegetation correction during the retrieval leave a characteristic fingerprint in the soil moisture time series. We find that time-variable offsets and sensitivities are commonly associated with an imperfect vegetation correction. Especially the changes in sensitivity can be large, with seasonal variations of up to 40%. Variations of this size impede the seasonal comparison of soil moisture dynamics and the detection of extreme events. Also, estimates of vegetation-hydrology coupling can be distorted, as the SMAP soil moisture has larger $R^2$ values with a biomass proxy than the in-situ data, whereas noise alone would induce the opposite effect. This observation highlights that time-variable biases can easily give rise to distorted results and misleading interpretations. They should hence be accounted for in observational and modelling studies.

## 1 Introduction

Soil moisture products derived from satellite measurements are subject to errors. These are not constant, but vary in space and time. For any given location, they may depend on variable factors such as vegetation phenology, atmospheric conditions or measurement characteristics like the incidence angle (Loew and Schlenz, 2011; Entekhabi et al., 2010; Su et al., 2016). Vegetation is commonly considered to be the most delicate factor to control when retrieving soil moisture from the raw satellite measurements (Dorigo et al., 2017; Chan et al., 2016). An imperfect vegetation correction during the retrieval will induce time-variable errors in the soil moisture product (Konings et al., 2017). As such structural errors potentially distort estimates

of vegetation-soil moisture coupling in unexpected ways (cf. Doherty and Welter, 2010), they are more pernicious than simple quasi-random noise or than time-constant biases.

However, the time-variable error properties of soil moisture products are poorly understood and rarely considered in practice (Loew and Schlenz, 2011). Arguably one reason for this is that there is currently no consistent way of estimating such rich error structures from data. If an exact reference soil moisture measurement were available, this would be an easy enterprise (Su and Ryu, 2015). Dense in-situ measurements are widely considered to be the closest one can get to perfect reference data, but they are rare, and non-negligible uncertainties remain (Colliander et al., 2017). In absence of perfect reference data, any useful error estimation procedure must cope with errors in all input data. Triple collocation and its various extensions can provide consistent error estimates in these circumstances (Gruber et al., 2016; Zwieback et al., 2016). However, similar to the standard RMSE metric they cannot directly separate non-constant systematic errors from quasi-random measurement noise. Conversely, common pre-processing steps (forming anomalies, analysis of short time periods) provide a simple means to deal with certain non-constant errors but lack generality (Loew and Schlenz, 2011; Gruber et al., 2016).

Here, we extend triple collocation to estimate non-constant error structures (Sec. 2). The central idea is to express systematic errors such as an offset and random errors in terms of explanatory variables like a vegetation index (cf. Xu et al., 2017; Doherty and Welter, 2010). The choice of explanatory variables should be informed by the measurement principle and the retrieval algorithm. In our case study of the SMAP product, we will specify it in terms of a misspecification of the vegetation correction during the retrieval process. We generally assume that three independent and noisy products are available. Once their hypothesized error structure has been specified, our Bayesian Triple Collocation approach proceeds in two steps. First, the specified error structure is embedded in a probabilistic model that links the unknown soil moisture $\theta$ with observed soil moisture products $y_n$. Second, Bayesian inference is applied, and one thus obtains estimates and uncertainties of the error parameters of interest. A simulation study indicates that time-variable systematic errors can be estimated reliably for as few as 250 samples (Sec. 3).

We apply this procedure to estimate time-variable biases in the SMAP soil moisture product that are associated with an imperfect vegetation correction (Sec. 4). This is not to say that other soil moisture products are not subject to similar biases; on the contrary, the SMAP baseline product is widely considered to provide the most reliable global soil moisture data record available (Chan et al., 2017; Colliander et al., 2017). To estimate soil moisture from the passive microwave measurements, the retrieval algorithm has to correct for the vegetation influence (Kurum et al., 2011; Chan et al., 2017). Specifically, the SMAP algorithm assumes that the vegetation optical depth $\tau$, a dimensionless measure of how much the microwaves interact with the vegetation, is known a priori.

We focus on croplands, as they present a particular challenge to the vegetation correction approach using an a priori $\tau$. Over crops, the input $\tau$, which is derived from the Normalized Difference Vegetation Index (NDVI), only provides an incomplete picture of the vegetation influence at microwave frequencies. The NDVI, being strongly influenced by leaf chemistry, can only indirectly account for the dominant controls on $\tau$, namely the vegetation water content and the canopy structure, both of which are particularly diverse and dynamic in crops (Lawrence et al., 2014; Momen et al., 2017). The NDVI-based input $\tau$ also does not account for inter-annual variability in vegetation conditions, which cannot be neglected over croplands (Patton and

Hornbuckle, 2013). Consequently, agricultural regions have been identified as a weak spot for the SMAP product (Colliander et al., 2017). However, the time-average metrics analysed so far cannot distinguish seasonal biases from quasi-random noise, and the magnitude of vegetation-induced systematic errors thus remains unknown.

We hypothesize that seasonal changes in the error structure arise due to an inaccurate vegetation correction in the retrieval, so that the biases relative to the in-situ data track the misspecification in the vegetation optical depth $\Delta\tau$. We specify the SMAP offset and sensitivity as a function of $\Delta\tau$, based on predictions by the $\tau$-$\omega$ radiative transfer model (Kurum et al., 2011). We estimate the associated error parameters with the Bayesian triple collocation approach, using in-situ and re-analysis data as additional soil moisture products. We find that SMAP soil moisture biases that track $\Delta\tau$ are widespread and large over croplands. This is especially so for the sensitivity, resulting in a time-dependent dynamic range of the SMAP soil moisture product that impedes seasonal comparisons of soil moisture dynamics. We attribute the time-variable biases to the imperfect vegetation correction, as the inferred bias characteristics largely match those predicted by the $\tau$-$\omega$ model. To illustrate the potential influence of these biases on estimates of vegetation-hydrology coupling (Adegoke and Carleton, 2002), we show that the coefficient of determination between SMOS $\tau$ anomalies and SMAP soil moisture anomalies is inflated compared to in-situ soil moisture measurements. In summary, our analyses suggest that soil moisture products can be subject to previously neglected time-variable biases that should be accounted for in observational and modelling studies.

## 2   Bayesian triple collocation

We now present a general overview of the approach (Fig. 1), which consists of two components. First, a probabilistic model that links the unknown soil moisture $\theta$ with the observed soil moisture products $y_n$. The link itself characterizes the error structure: it depends on error parameters $\gamma_{y_n}$ and explanatory variables $w$. Second, a Bayesian inference approach that provides estimates of all the unknown quantities, in particular the error parameters. By conditioning on the observed soil moisture data (input), we get a posterior distribution over the unobservable quantities (output). We focus on a setting inspired by triple collocation studies, i.e. we for the most part assume that $N = 3$ independent and noisy products are available (Gruber et al., 2016). In regular triple collocation, three independent products provide sufficient information to estimate the $\sigma$ of all three products and bias parameters of two of the three products. In a Bayesian setting, the presence of prior information allows one to reduce the number of independent products, but the results will strongly depend on the prior distributions.

Our approach has several characteristics that make it useful in a wide range of applications. It is widely applicable, as no soil moisture product is assumed to be free of errors. This is particularly critical for estimating the noise magnitude and the sensitivity, which cannot be estimated consistently by standard regression approaches when the reference product is subject to errors (Yilmaz and Crow, 2013). Also, it provides principled uncertainty estimates through the posterior distribution. It is flexible, as it can be adapted to many functional relations and error structures. Finally, it is transparent because all modelling assumptions are explicit. Owing to its flexibility, the model can be modified to test the sensitivity of the results to certain assumptions. We will exploit these advantages by modifying the prior distribution and the likelihood; for instance, we will test several models for the unknown soil moisture $\theta$.

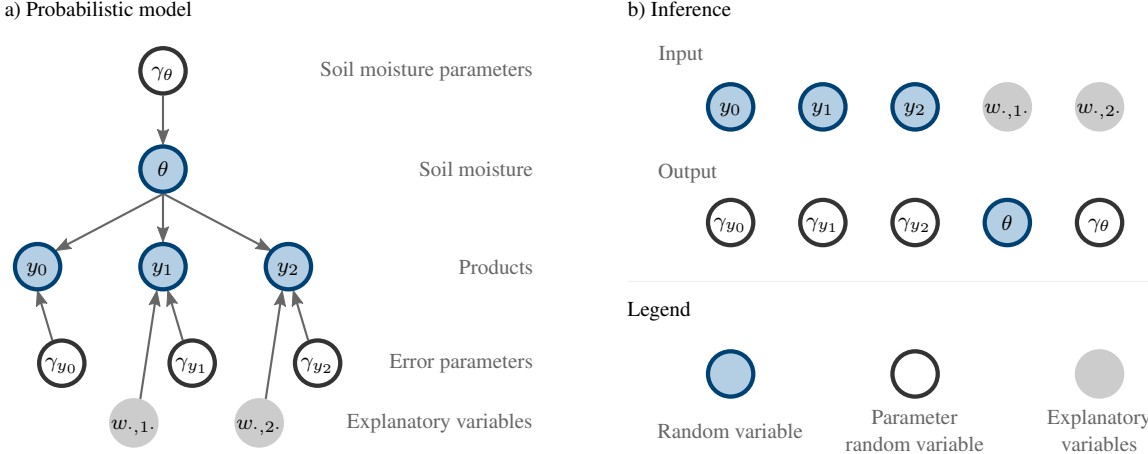

**Figure 1.** The two components of Bayesian triple collocation. a) The probabilistic model expresses the observable products in terms of the unknown soil moisture, error parameters and explanatory variables. The set of error parameters of the reference product, $\gamma_{y_0}$, does not contain variable bias terms. b) The soil moisture products and explanatory variables serve as input to the inference, which produces estimates (posterior probability distributions) of, amongst other things, the error parameters.

## 2.1 Probabilistic model

The probability distribution comprises the observable products $y_0$ to $y_{N-1}$, the unknown soil moisture $\theta$ and numerous parameters: the set of parameters $\gamma_{y_n}$ characterizes product $n$, and $\gamma_\theta$ does the same for the soil moisture. The structure of the model is summarized in Fig. 1a, which depicts the corresponding directed acyclic graph (MacKay, 2003). This graph expresses how

the distribution over all the random variables is factorized into smaller components. Starting with an observable product $y_n$, its distribution is modelled conditional on the unknown $\theta$, the associated parameters $\gamma_{y_n}$ (including the time-independent and time-dependent bias parameters) and the explanatory variables $w_{\cdot,n}$. We refer to this conditional distribution as the error model. It is conditional on the soil moisture $\theta$, whose distribution (conditional on parameters $\gamma_\theta$) is referred to as the soil moisture model. The final component are the parameters $\gamma_{y_n}$ and $\gamma_\theta$, which are assigned prior distributions. These prior distributions are

integral to the Bayesian approach, and they express the initial belief about the parameters' likely values. We now describe each of these components in turn. To facilitate future applications of the approach, we do this using very general notation, which we will later in the SMAP case study simplify by dropping subscripts.

### 2.1.1 Error model

Each product's error model is governed by a set of parameters $\gamma_{y_n}$ which quantify the error component such as the biases

(systematic errors). We consider an affine error model according to which the dynamic range or sensitivity of the product can differ from that of the true soil moisture, governed by the scaling parameter $L_n(t)$. We also include an additive offset or bias

$M_n(t)$ (Zwieback et al., 2012; Yilmaz and Crow, 2013), yielding

$$y_n(t) = L_n(t)\left(\theta(t) - \theta_0\right) + \left(\theta_0 + M_n(t)\right) + \epsilon_n(t)\,, \tag{1}$$

where we essentially relate the deviation of soil moisture from a typical, prescribed soil moisture $\theta_0$ to the observed product.

Our key extension compared to previous triple collocation studies is to make the bias terms and the noise magnitude vary with time-dependent explanatory variables, $w_{\cdot,np}(t)$:

$$L_n(t) = l_n + \sum_{p=1}^{P_{\lambda,n}} \lambda_{np} w_{\lambda,np}(t) \tag{2}$$

$$M_n(t) = m_n + \sum_{p=1}^{P_{\mu,n}} \mu_{np} w_{\mu,np}(t) \tag{3}$$

where $l_n$ and $m_n$ are time-invariant values specific to product $n$, and the parameters $\lambda_{np}$ and $\mu_{np}$ are coefficients that quantify the dependence on the $p$th explanatory variable $w_{\lambda,np}$ and $w_{\mu,np}$, respectively. The explanatory variables can depend on the product $n$ as well as on the parameter $(\mu, \lambda)$. They are external quantities that are part of the model input. To facilitate the interpretation of the parameters, we always assume the explanatory variables to have zero mean and unit standard deviation (Gelman et al., 2008). $l_n$ and $m_n$ thus represent the time-average biases. The magnitude of $\lambda_n$ and $\mu_n$ represents the magnitude of the associated temporal changes in $L_n$ and $M_n$, respectively.

The quasi-random errors are characterized by their variance and further distributional assumptions. For the variance $S_n^2(t) = E(\epsilon_n(t)^2)$ we suggest a multiplicative model that is commonly employed in regression studies (Harvey, 1976)

$$S_n^2(t) = \sigma_n^2(t) \prod_{p=1}^{P_{\sigma,n}} w_{\sigma,np}(t)^{\kappa_{np}} \tag{4}$$

where $\kappa$ governs the sensitivity of the error variance to the explanatory variable, which has to be positive. We always assume it to be normalized so its geometric mean is one, as this simplifies the interpretation of the typical variance $\sigma_n^2(t)$. A positive value $\kappa > 0$ indicates a larger variance as the explanatory variable increases, and a negative value corresponds to a smaller variance. To complete the specification of the errors, a probability distribution has to be assumed. We focus on a normal distribution with zero mean and variance given by $S_n^2$, as our robustness checks indicated that the results were not very sensitive to this assumption (cf. Sec. 3). Finally, two further properties have to be specified. First, we assume that the errors at different times are independent. We will later show that violations of this assumption commonly have negligible impact on the estimated parameter values. Second, we postulate that the errors of the various products are independent. This is generally a key assumption in triple collocation-type studies (Gruber et al., 2016).

We generally specify $y_0$ to be the reference product (Yilmaz and Crow, 2013; Gruber et al., 2016). Its error magnitude is assumed to be constant over time, and so are its additive bias ($M = 0\,\mathrm{m^3\,m^{-3}}$) and its sensitivity $L = 1$. This constraint ensures that the unknown soil moisture distribution (its mean and scale) can be inferred from data, as it essentially specified what the reference product was a noisy estimate of (Zwieback et al., 2016). The bias terms are thus always relative to this reference product.

### 2.1.2 Soil moisture model

The second piece of the probabilistic model concerns the soil moisture $\theta$, the distribution of which also has to be specified. Our default representation is a simple logistic model. The physical quantity $\theta$, which is constrained to between 0 and the porosity $\phi$, is expressed as a function of the non-dimensional unbounded soil moisture $\Theta$

$$\theta(t) = \phi \frac{1}{1 + \exp(-A - B\Theta(t))} \text{ with } \Theta \sim \mathcal{N}(0,1). \tag{5}$$

$\Theta$ is modelled as a standard normal random variable. The site-specific parameters $A$, $B$ and $\phi$ are inferred in the Bayesian inference. We summarized the parameters of the soil moisture model under $\gamma_\theta$.

One drawback of this model is that it cannot account for the autocorrelation and seasonality of soil moisture. To test for the importance of temporal characteristics, we also generalize the model by making $A$ and $B$ time dependent. We do this by expanding $A$ and $B$ in a spline basis with 12 monthly basis functions. While this model cannot completely account for the complex temporal characteristics of soil moisture, it captures the seasonal trends.

### 2.1.3 Prior distributions

To complete the full probability distribution, one has to specify the prior distributions of the parameters $\gamma_\theta$ and $\gamma_{y_n}$. As we work with normalized explanatory variables, we can use a problem-independent prior distribution (Gelman et al., 2008). We choose the priors to be weakly informative, thus partially ruling out unreasonable values but still letting the data speak for themselves (Gelman et al., 2008). Our default choices are summarized in Tab. 1.

For all products $y_n$, we put a very weak prior on the error magnitude variance $\sigma^2$. It is given by an exponential distribution with mean 0.1 $(\mathrm{m}^3\,\mathrm{m}^{-3})^2$, plotted in Fig. S1. In other words, we barely constrain this quantity. For the product error parameters, $m$ and $\mu$ [$\mathrm{m}^3\,\mathrm{m}^{-3}$] are assumed distributed according to a t distribution $T(0, 0.3^2; 4)$; Fig. S1. It is centred at 0 with a standard deviation of 0.3 and very heavy tails due to its four degrees of freedom. These values barely constrain the estimation of the additive bias $m$, as they do not rule out biases as large as 0.5 $\mathrm{m}^3\,\mathrm{m}^{-3}$. Similarly for $l$ and $\lambda$ [-], whose prior is $T(1, 0.3^2; 4)$.

The standard prior distribution for the soil moisture parameters $\gamma_\theta$ follows a similar logic. The porosity [$\mathrm{m}^3\,\mathrm{m}^{-3}$] is given by $T(0.4, 0.1^2; 4)$. The parameters $A$ and $B$ in the standard logistic model are assigned $T(0, 3.0^2; 4)$ and an exponential distribution with mean 3.0, respectively.

## 2.2 Bayesian inference

The Bayesian inference takes the observed products and explanatory variables as input and outputs posterior probability distributions over the unknown quantities (Fig. 1). The posterior distributions are obtained from the probabilistic model by conditioning on the input data. Conditioning is a well-defined mathematical operation, but analytical solutions are infeasible for complicated models like ours (MacKay, 2003). Instead, one has to resort to approximations. Monte Carlo methods are arguably the most popular. Their output is an approximation to the posterior distribution that consists of samples drawn from this distribution. Here, we rely on Hamiltonian Monte Carlo as implemented using the adaptive No-U-Turn Sampler in pymc3 (Hoffman

**Table 1.** The default model specification used in both the simulation study and the SMAP case study, and the baseline configuration for the simulation runs. The bias terms for the reference product $y_0$ were assumed known.

|  | Model specification |  | Simulation parameters |
|---|---|---|---|
| **Errors** |  |  |  |
| $\sigma$ [m$^3$ m$^{-3}$] | exponential prior |  | [0.02, 0.04, 0.05] |
| $m$ [m$^3$ m$^{-3}$] | Student prior | $m_0 = 0$ | [0.00, 0.03, -0.05] |
| $l$ [-] | Student prior | $l_0 = 1$ | [1.0, 1.1, 0.9] |
| $\mu$ [m$^3$ m$^{-3}$] | Student prior | $\mu_0 = 0$ | [0.0, 0.02, -0.02] |
| $\lambda$ [-] | Student prior | $\lambda_0 = 0$ | [0.0, 0.06, 0.0] |
| $\kappa$ [-] | Student prior | $\kappa_0 = 0$ | [0.0, 0.2, -0.2] |
| **Ancillary components** |  |  |  |
| $\epsilon$ distribution | Normal |  | Normal |
| $\theta$ distribution | logistic ($A$, $B$, $\phi$) |  | logistic |

and Gelman, 2014; Salvatier et al., 2016). The No-U-Turn Sampler produces successive, dependent samples of the posterior distribution that are called a chain. Each sample consists of draws from the posterior distribution, or actually an approximation thereof, of all the unobserved random variables (Output in Fig. 1b). They comprise the parameter random variables (e.g. the time-dependent biases) as well as the soil moisture time series, i.e. one value of $\theta$ for each SMAP observation. For each location, we sample two independent chains with 2000 samples each, which standard quality controls (divergences, chain mixing) indicate is sufficient. Following common practice, the first 1000 samples are discarded (Brooks and Gelman, 1998)

## 3 Simulation study

We now study the applicability of Bayesian triple collocation using a simulation study. We used three simulated products with realistic error properties (Tab. 1). $y_0$ was taken to be the reference product in both the simulation and in the probabilistic models. For the other two products the biases and error magnitudes were assumed dependent on a normalized explanatory variable that varied seasonally (Tab. 1). The soil moisture was prescribed using a simple antecedent precipitation model driven by a seasonally-dependent Hidden Markov Model rainfall generator, which gave rise to autocorrelated soil moisture. Mimicking the SMAP sensor, an observation was made every 2-4 days.

We first analysed the fidelity with which the error parameters could be estimated. To this end, we simulated $R = 25$ time series and computed the posterior distribution using the default probability model of the previous section, summarized in Tab. 1. We computed an aggregated RMSE error for each parameter $\pi$ by comparing the prescribed parameters $\pi_n$ for all products

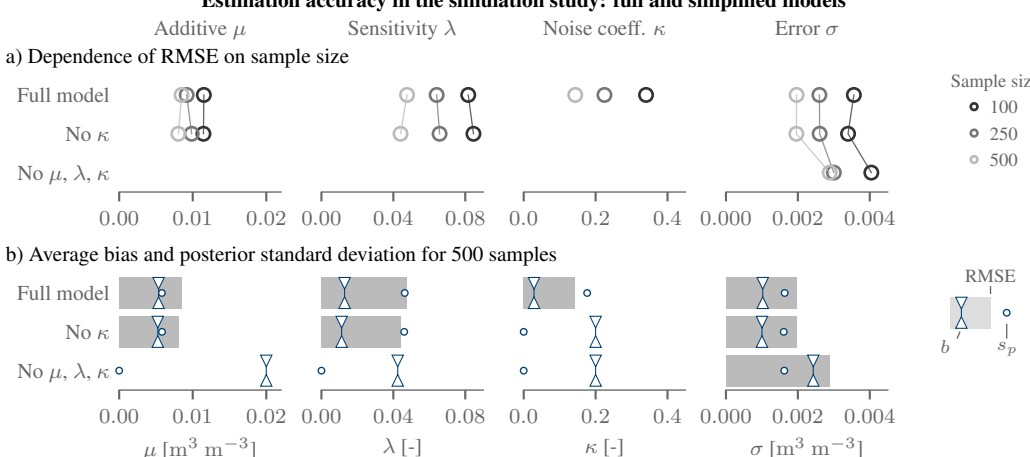

**Figure 2.** Simulation results illustrating the estimation fidelity. a) Dependence on sample size. b) Comparison of RMSE and the simulation bias magnitude $b$ to the posterior standard deviation $s_p$. In a) and b) the first line shows the full model, whereas $\kappa$ is set to zero in the inference in the second, and $\lambda$, $\mu$ and $\kappa$ were set to zero in the third. In these cases, the RMSE is equal to $b$, and these values are not shown in a).

with the posterior means inferred from each simulation run $r$

$$\text{RMSE} = \left( \frac{1}{N_\pi R} \sum_{n=1}^{N_\pi} \sum_{r=1}^{R} (\hat{\pi}_{n,r} - \pi_n)^2 \right)^{\frac{1}{2}} \tag{6}$$

where $\hat{\pi}_{n,r}$ is the posterior mean of parameter $\pi$ estimated for product $n$ (out of $N_\pi$) in run $r$. To quantify systematic deviations, we computed an average absolute bias (in the root mean square sense) between the truth and posterior mean for each parameter:

$$5 \quad b = \left( \frac{1}{N_\pi R} \sum_{n=1}^{N_\pi} \left( \sum_{r=1}^{R} (\hat{\pi}_{n,r} - \pi_n)^2 \right) \right)^{\frac{1}{2}}, \tag{7}$$

and we subsequently averaged the magnitudes over all products (referred to as mean absolute bias). To explore the impact of the number of observations on the accuracy, we did this for 100, 250 and 500 observations, which bound the number of observations that were available in the SMAP study.

The error parameters could be estimated with sufficient fidelity in the simulation study (Fig. 2; full model). The accuracy
10 of the estimated time-variable additive bias parameter $\mu$ was found better than $0.01 \text{ m}^3 \text{ m}^{-3}$, and thus likely sufficient to detect relevant non-constant biases. The sensitivity coefficients $\lambda$ are retrieved with comparable precision: the RMSE of 0.05 corresponds to a differential bias between dry and wet conditions of around $0.01 \text{ m}^3 \text{ m}^{-3}$ (assuming a soil moisture dynamic range $\sim 0.25 \text{ m}^3 \text{ m}^{-3}$). Again, this should be sufficient for many applications. Finally, the time-constant noise magnitude $\sigma$ could be estimated accurately ($\ll 0.005 \text{ m}^3 \text{ m}^{-3}$).

15 Bayesian triple collocation yields a distribution of the parameters and thus naturally provides uncertainty estimates. The posterior standard deviation $s_p$ compares favourably to the RMSE errors: Fig. 2b shows that the posterior standard deviations

are within 10 to 20 % of the RMSE errors. Even though these two quantities represent different kinds of uncertainty, they are expected to be comparable for large samples (MacKay, 2003). The posterior distribution hence provides a useful summary of the estimation uncertainty.

To test the sensitivity of the estimates to model assumptions, we extended the simulation study. The most critical aspect turned out to be the specification of the bias terms: neglecting variable bias terms can impair the overall estimation quality. Neglecting the complex error structure leads to an overestimation of the error magnitude (Fig. 2). In our case, setting $\mu$, $\lambda$ and $\kappa$ to 0 induced an increase in the RMSE of $\sigma$ by a factor of two. This increase was mainly due to a large bias, as the varying offsets were wrongly attributed to quasi-random noise. The posterior uncertainty estimates also became inaccurate. Conversely, neglecting only the variability of the error magnitude (i.e. setting $\kappa = 0$) had limited impact on the retrieval of the other parameters.

To test for additional model assumptions, we modified the model and the forward simulations. The impact on the estimation accuracy was typically limited (see the supplement), so we only provide a short summary. First, the model for the noise term $\epsilon$ had a moderate influence on the estimation quality. A mismatch between assumed and simulated $\epsilon$ distributions increased the RMSE of the error variance by a small amount (<0.001 $\mathrm{m}^3\,\mathrm{m}^{-3}$ for $\sigma$) for a range of error distributions and strongly autocorrelated errors. The other crucial assumption in the model is the probability distribution for the soil moisture. Also here, the changes are typically small (up to 10% improvement in the RMSE, but a decrease in bias) when replacing the standard time-invariant model by a seasonally variable model or by a different time-invariant model. The improvement suggests that the model-internal soil moisture distribution can have an impact on the estimated biases, in particular when the actual soil moisture is correlated with the explanatory variable, as it was in the simulated data. We would hence expect that for most applications it is the seasonal and sub-seasonal time scales that the soil moisture model should be able to capture. For comparison, autocorrelation on the inter-storm time scale that is not captured by our model but present in the simulated data did not seem to introduce major limitations (sufficient fidelity for the full model, Fig. 2). The prior distributions had an even smaller impact. Making the prior distributions twice as wide or the tails less heavy changed the estimates by only a few percent.

## 4 SMAP case study

### 4.1 Materials and methods

#### 4.1.1 Data

To estimate the biases of the SMAP soil moisture product, we used $N = 3$ soil moisture data sets in the probabilistic inference: apart from the SMAP product, these were in-situ data from dense networks or sparse sites, and MERRA-2 reanalysis data. The SMAP data set we analysed was the SMAP enhanced Level-2 soil moisture product, which is disseminated on the 9 km EASE-Grid 2.0 at a resolution of 33 km (Chan et al., 2017; O' Neill et al., 2017). It contains a variety of estimates of the top (5 cm) soil moisture, of which we chose the standard product (baseline V single channel algorithm, 9 am morning passes). The single channel retrievals rely on an a priori $\tau$ derived from a MODIS climatology; these $\tau$ values are included in the disseminated

**Table 2.** Network sites from north to south, including their Koeppen-Geiger climate regime.

| Site | Location | Climate | Crop cover [%] | Samples |
|------|----------|---------|---------------:|--------:|
| Kenaston | Canada (Saskatchewan) | Cold | 90 | 351 |
| Carman | Canada (Manitoba) | Cold | 80 | 352 |
| South Fork | USA (Iowa) | Cold | 90 | 323 |
| REMEDHUS | Spain | Temperate | 80 | 475 |
| Fort Cobb | USA (Oklahoma) | Temperate | 60 | 413 |
| Bell Ville | Argentina | Arid | 90 | 399 |
| Monte Buey | Argentina | Arid | 90 | 400 |

product. We studied all available data since the beginning of the record in April 2015 until August 2017, i.e. up to three annual growing seasons. After removing flagged retrievals (Colliander et al., 2017), the number of available measurements is on the order of 300-500, which is not ideal but should be sufficient according to Fig. 2a.

The analyses focus on seven locations in North America, South America and Europe with significant crop cover, due to the availability of high-quality dense in-situ networks (Tab. 2). At these SMAP core or candidate sites, continuous calibrated in-situ measurements at 5 cm depth are collected at multiple locations within a SMAP grid cell (Colliander, 2017; Colliander et al., 2017).

To provide a better overview of the spatial patterns, we also used data from > 200 in-situ sites in the contiguous United States (SCAN and USCRN networks). These sparse sites consist of a single station per satellite pixel, and their representativeness is hence not comparable to that of the network sites. The USDA's SCAN network has been in continuous operation since 1999 and provides soil moisture data at 2 in (5 cm) depth (Schaefer et al., 2007). The USCRN network consists of 114 sites whose location was chosen to be maximally representative of its surroundings; we used the 5 cm soil moisture observations (Bell et al., 2013; Diamond et al., 2013; Palecki et al., 2017). We assigned these sites a dominant land cover based on the MODIS MCD12C1 land cover product (Friedl et al., 2017).

For the third soil moisture product we used the MERRA2 reanalysis (M2T1NXLND.5.12.4) (Gelaro et al., 2017; Global Modeling and Assimilation Office, 2017). It is the most recent reanalysis product of NASA's Global Modeling Office, available at a resolution of 55 km. For sensitivity analyses, we also used GLDAS-2 (Noah model, GLDAS_NOAH025_3H.2.1), a popular land assimilation data set (Roddell and Beaudoing, 2017).

To quantify the error structure as a function of $\Delta\tau$, we estimated $\Delta\tau$ as the difference between an external estimate of $\tau$ and the SMAP input $\tau$. The external estimate was based on independent L-band microwave observations by the SMOS satellite (Level 3 operational algorithm). The multi-angular observations and the temporal aggregation of multiple overpasses are conducive to providing robust, if noisy, measurements of the vegetation optical depth relevant to SMAP (Al Bitar et al., 2017). To reduce the impact of high-frequency noise, the $\Delta\tau$ data were smoothed to a temporal resolution of 16 days (LOWESS

filter). The explanatory variable used in the Bayesian inference is the normalized version

$$w_{\Delta\tau}(t) = \frac{1}{\text{std}(\Delta\tau)} \left( \Delta\tau(t) - \text{mean}(\Delta\tau) \right).$$
(8)

### 4.1.2 Error estimation

In our estimation we specified the error structure based on our hypothesized impact of a vegetation misspecification. The $\tau$-$\omega$
model (Kurum et al., 2011) predicts that a vegetation misspecification $\Delta\tau = \tau_{\text{inv}} - \tau_{\text{true}}$ will induce both a sensitivity $L$ and an
offset $M$ which scale approximately linearly with $\Delta\tau$ (Fig. 3a). There is thus only one explanatory variable ($P_{\lambda,n} = P_{\mu,n} = 1$),
namely the normalized $w_{\Delta\tau}$ of Eq. 8. By slightly simplifying the notation, the error model reads

$$y_{\text{SMAP}}(t) = \underbrace{(l + \lambda w_{\Delta\tau}(t))}_{L(t)} (\theta(t) - \theta_0)$$
$$+ \underbrace{(m + \mu w_{\Delta\tau}(t))}_{M(t)} + \theta_0 + \epsilon(t).$$
(9)

Here $\theta_0$ was taken to be the mean value of the in-situ product. We quickly recall the interpretation of the error parameters.
$\lambda$ and $\mu$ quantify the temporal changes in the sensitivity and offset, respectively, that are associated with changes in $\Delta\tau$.
Their magnitudes correspond to the temporal standard deviation of the sensitivity and offset, respectively (Fig. 3b). Their
sign expresses the direction of the dependence on $\Delta\tau$. It is predicted to be positive: as $\Delta\tau$ grows, the inversion increasingly
overcompensates the vegetation-induced loss of sensitivity to soil moisture and increase in brightness temperature, which in
turn inflates both $L$ and $M$, respectively.

To compute the predicted biases in Fig. 3a), we assumed the $\tau$-$\omega$ model applied and was correctly specified (temperature,
dielectric mixing model [Dobson; silt loam], single-scattering albedo $\omega = 0.05$, etc.). For a given value of $\tau_{\text{true}}$, we simulated
the V-polarized brightness temperatures for dry and wet soil moisture conditions. These brightness temperatures were in turn
the basis for estimating soil moisture by inverting the $\tau$-$\omega$ model using the wrong $\tau_{\text{inv}}$ as a function of $\Delta\tau$. For both dry and
wet soil moisture conditions, the deviation was an estimate of the retrieval bias: their mean was taken to be an estimate of
the offset $M$, whereas their difference allowed us to estimate $L$. When plotted against $\Delta\tau$, $M$ and $L$ increase nearly linearly
and only show a weak dependence on $t\tau_{\text{true}}$. The slope of this relation is thus well but not perfectly defined. We refer to the
slopes as $\mu^\star$ (for $M$) and $\lambda^\star$ (for $L$), respectively. To account for the spread due to the slight curvature and dependence on $\tau$,
we estimated the likely range of values by computing the slopes from the differences in $L$ or $M$ between five equally spaced
values of $\Delta\tau$ (between -0.1 and 0.1), repeated for equally many values of $\tau_{\text{true}}$ (between 0.1 and 0.6). The range of these
values was $\lambda^\star_{\text{pred}} \in [2.0, 3.8]$ and $\mu^\star_{\text{pred}} \in [0.33, 0.65]$ m$^3$ m$^{-3}$. These ranges will later be compared to data-driven estimates,
thus providing a first-order assessment of the agreement between predictions and observations, despite the simplified setup and
neglect of other retrieval errors.

The same overcompensation that increases the sensitivity also increases the noise level, so that we also made the variance of
$\epsilon(t)$, $S^2(t)$, time dependent. We used two explanatory variables, $\exp\Delta\tau$ and the external $\tau$ (both normalized, $P_{\kappa,n} = 2$). The
reason for also including $\tau$ was that the noise level $S^2$ is predicted to depend on $\tau$ even if $\Delta\tau$ is constant, in contrast to the bias

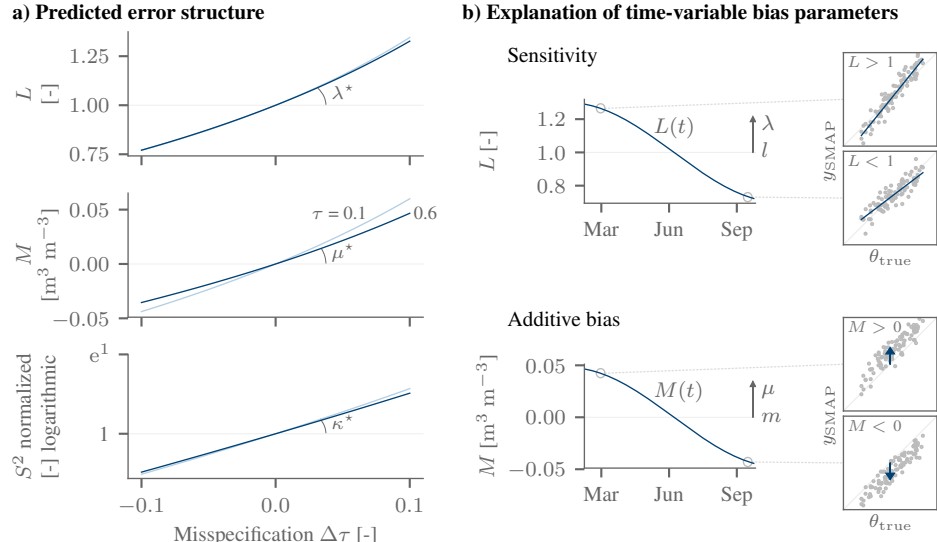

**Figure 3.** a) The expected error structure due to a misspecified vegetation in the SMAP retrieval. Additive bias $M$, sensitivity $L$ and variable noise level $S^2$ induced by a misspecified vegetation optical depth $\tau$, as predicted by the $\tau$-$\omega$ model for the SMAP satellite and single-scattering albedo $\omega = 0.05$. Two different values of the true $\tau$ were assumed (0.06 in dark blue, 0.01 in light blue). The slopes of these approximately linear relationships will later be compared to independent data-driven estimates. b) General explanation of the bias terms, illustrated for a hypothetical time-changing sensitivity $L(t)$ and offset $M(t)$ (underlying change in $\Delta\tau$ not shown). A varying sensitivity changes the response of the SMAP retrieval to a unit change in the true soil moisture. When it is larger than one, the SMAP data have a larger dynamic range than the true soil moisture (illustrated by the slope > 1 in the inset). The time-average value of $L$ is $l$, and the temporal standard deviation of $L$ is given by $|\lambda|$ (length of arrow). A variable $M$ induces non-constant offsets, and the magnitude of its temporal variability is given by $|\mu|$. $M > 0$ corresponds to a positive offset (shown in the inset).

terms. By transforming $\Delta\tau$ to $\exp\Delta\tau$, the predicted dependence of $S^2$ shown in Fig. 3 could be reproduced accurately. The dependence on $\exp\Delta\tau$ is positive, i.e. an increase in $\Delta\tau$ is associated with an increase in noise level. We would thus expect $\kappa_{\exp\Delta\tau} > 0$.

The same error structure was assumed for the re-analysis data. The inclusion of a $\Delta\tau$-dependent bias for the reanalysis product is not driven by physical reasoning, but for statistical reasons. By controlling for the same explanatory variables for both products, the impact of potential confounders - e.g. a seasonal bias that is correlated with $\Delta\tau$ - on the bias estimates of the remotely sensed product can be reduced. If this were not done, the model would try to partially adjust the time-variable bias term of the remote sensing product to minimize the systematic differences to the re-analysis product, thus distorting the bias estimates. The in-situ observations were taken as reference product $y_0$: $L = 1$, $M = 0$, $\kappa = 0$). All other model components – the soil moisture and error distributions, the priors – were identical to the simulation study (Tab. 1).

To compare the estimated biases with the model predictions, we re-expressed $\lambda$ and $\mu$ in absolute terms by reversing the rescaling from $\Delta\tau$ to $w_{\Delta\tau}$. Thus,

$$\lambda^\star = \frac{\lambda}{\text{std}(\Delta\tau)} \quad \text{and} \quad \mu^\star = \frac{\mu}{\text{std}(\Delta\tau)} \tag{10}$$

are, respectively, the slope of $L$ and $M$ vs $\Delta\tau$. In other words, the estimated $\lambda^\star$ describes the inferred change in $L$ for a unit

change in $\Delta\tau$, and similarly for $\mu^\star$. They can be directly compared to the model predictions described above. Note that the division only reverses the scaling but not the offset inherent in the normalization from $\Delta\tau$ to $w_{\Delta\tau}$. Estimating the derivative at the mean value of $\Delta\tau$ (due to the offset) rather than at $\Delta\tau = 0$ has, however, no impact for a purely linear relation.

To test the robustness of the estimates, we varied the input data and the model configuration. Instead of using the SMOS $\tau$ as reference, we also derived $\Delta\tau$ from the SMAP dual channel (LOWESS smoothed) retrievals (O' Neill et al., 2017) and from

10 contemporaneous MODIS NDVI data (Didan, 2017). We converted the MODIS NDVI to $\tau$ using the same equations as in the generation of the SMAP input climatology. We refer to these model runs as SMAP DC $\tau$ and MODIS $\tau$, respectively. We also modified the external soil moisture products: instead of MERRA-2 we used GLDAS-2 (GLDAS $\theta$), and we also dropped the reanalysis data set altogether (no reanalysis). This is possible in a Bayesian setting; to improve the identification of the errors, we assigned a narrower prior distribution to the in-situ noise magnitude ($0.02\ \text{m}^3\,\text{m}^{-3}$). To account for a potential confounding

of $\tau$ itself, which may also have an effect on the biases, we included the smoothed SMOS $\tau$ as second explanatory variable for $L$ and $M$, referred to as $\tau$ control. Finally, we also modified the model configuration. We made the probability model for $\theta$ vary seasonally ($\theta$ spline) to account for the non-negligible impact of this detail of the model specification on the estimates observed in the simulation (Sec. 2.1.2).

### 4.1.3 Estimates of vegetation-soil moisture coupling

To explore the relation between time-variable vegetation biases and estimates of vegetation-water coupling, we analysed the coefficient of determination $R^2$ between $\tau$ and soil moisture anomalies. These were derived from the SMOS $\tau$ time series and both SMAP and in-situ soil moisture, respectively. The associated anomalies $\tau'$ and $\theta'$ were obtained by subtracting a seasonal climatology that in turn was the smoothed (30 day) multi-year average of the input time series. To compare the SMAP and the in-situ soil moisture data, we then computed the difference in the coefficient of determination $\Delta R^2$

$$\Delta R^2 = R^2(\tau'_{\text{SMOS}}, \theta'_{\text{SMAP}}) - R^2(\tau'_{\text{SMOS}}, \theta'_{\text{in}-\text{situ}}) \tag{11}$$

If the SMAP soil moisture were only contaminated by random noise relative to the in-situ data, $\Delta R^2$ would tend to be negative. Time-variable vegetation biases, on the other hand, can induce positive values. Time-average biases cancel out. Of course, there are also other error sources such as representativeness errors, especially for the sparse networks. Further, the time series are comparatively short, but $\Delta R^2$ can provide a first tentative assessment of the reliability of coupling metrics such as $R^2$ in the

30 presence of time-variable biases.

## 4.2 Results

### 4.2.1 Network sites

SMAP biases that track the vegetation misspecification $\Delta\tau$ are common at the core validation sites. Especially changes in the sensitivity can be large. Reanalysis bias parameters were estimated as well, but they are considerably smaller in magnitude than those of the SMAP product.

The varying sensitivity of SMAP is illustrated for the South Fork site, Iowa (USA), in Fig. 4a. In July, when the predominantly cultivated corn is heading and flowering (Tomer et al., 2008), the magnitude of the SMAP response to rainfall events matches that of the in-situ data. Conversely, in autumn (senescence, post harvest) the sensitivity of SMAP to soil moisture variations is diminished. The Bayesian inference reproduces this visually inferred pattern, as the sensitivity $L$ drops by more than 0.5 (or 50%). By design, the temporal changes in $L$ are governed by changes in $\Delta\tau$, which drops by about 0.1 from July to September. Over the entire time series, its temporal variability is $|\lambda| = 0.3$. Note that even when $\Delta\tau \approx 0$ (August), the sensitivity is too low. As $\Delta\tau$ is essentially zero on average over the entire time series (-0.01), the sensitivity at $\Delta\tau = 0$ corresponds to the time-average value $l$. Its posterior median is 0.64 and thus less than the expected value of 1. We will return to the time-average biases later.

Pronounced changes in the sensitivity are found for all network sites but one (Fig. 4b). For the most part their magnitude is large, as the $|\lambda|$ values correspond to a temporal variability of around 10 to 50%. The inferred relation to $\Delta\tau$ is consistently positive ($\lambda > 0$), i.e. as $\Delta\tau$ increases over time, so does the SMAP sensitivity $L(t)$. The inferred direction hence matches the model prediction (Fig. 3a) of a positive $\lambda$. The model does not constrain the magnitude of the $\lambda$ parameter because the latter is an internally normalized quantity. When converted into absolute quantities ($\lambda^\star$), the inferred dependence of $L$ on $\Delta\tau$ matches the model predictions reasonably well (Fig. 4b). In other words, the data-derived, completely independent estimate is broadly consistent with the predicted impact of a $\tau$ misspecification in the retrieval, despite limitations in the estimates (e.g. issues with the reference $\tau$) and the model predictions (e.g. assumed knowledge of the land surface temperature) of $\lambda^\star$. There is no clear apparent dependence of $\lambda^\star$ on location or land cover properties; for instance, Monte Buey and Bell Ville are within $< 100$ km of one another, and despite the similarity in planted crops the latter's $\lambda^\star$ is considerably larger.

Also the additive biases track changes in $\Delta\tau$ at the network sites, but to a lesser extent (Fig. 4c). Over time, the biases vary by up to 0.02 m$^3$ m$^{-3}$ in magnitude. The inferred direction, $\mu > 0$, matches the predictions, as an increase in $\Delta\tau$ corresponds to a larger offset. However, the magnitude of the inferred change of the offset with $\Delta\tau$, $\mu^\star$, is generally smaller than predicted. A very small value of $\mu^\star$ is found at the South Fork site: it is a factor of ten smaller than its predicted range (Fig. 4c). It is also small enough to be practically irrelevant, as the additive bias at South Fork is inferred to be essentially constant (Fig. 4a).

The time-variable biases are complemented by the time-average biases, which are quite large at several network sites. The time-average sensitivity $l$ deviates from its nominal value of one by more than 30% at three out of 8 sites (Fig. 5). It tends to be larger than one, as may be expected considering the surface soil moisture SMAP is sensitive to has a larger dynamic range than the moisture at the depth of the probes. The South Fork site of (Fig. 4a) with $l < 1$ is thus somewhat unusual. However, it is typical in that its time-average additive bias $m$ is negative. Conversely, the noise level $\sigma$ inferred using our approach tends to

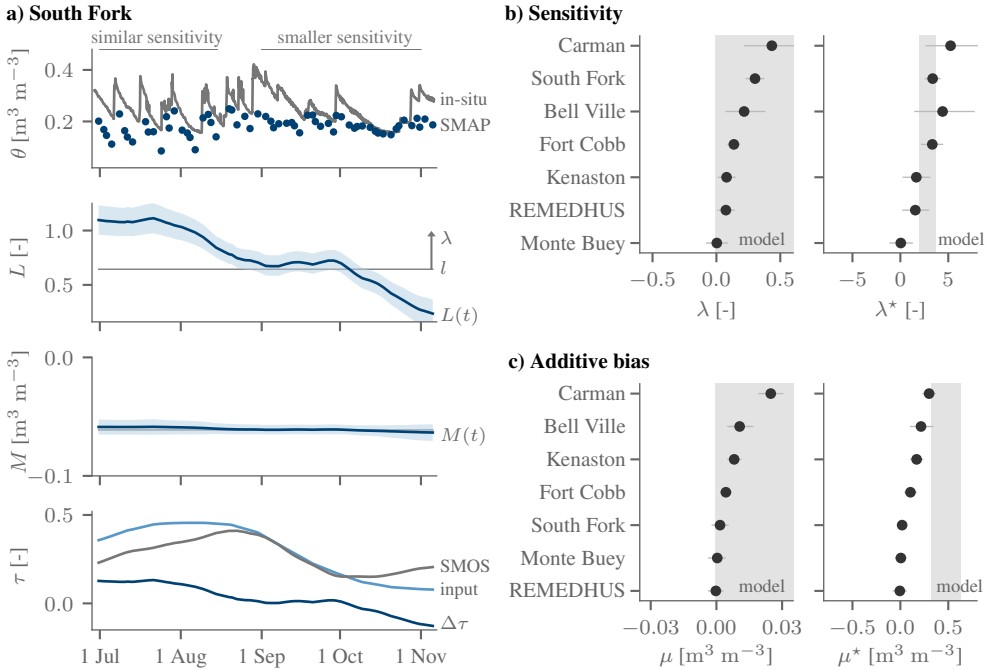

**Figure 4.** Time-variable biases over the network sites. a) The SMAP product's sensitivity decreases from summer to late autumn 2015 at the South Fork site (42.4N, 93.4W), which is reflected by a decrease in the inferred sensitivity $L$ that tracks a decrease in $\Delta\tau$: in early summer $L(t) \approx 1$, but it subsequently drops well below the time-average value of $l = 0.7$. Its temporal standard deviation is given by $\lambda = 0.3$ (arrow). $M(t)$ is approximately constant ($M(t) \approx m$) because $\mu$ is small (not shown). b) Over the network sites, the association of changes in the sensitivity with changes in $\Delta\tau$ is predominantly positive (marker: posterior median, lines: 5–95% uncertainty), as predicted by the model (positive $\lambda$; grey background). The magnitude of the dependence for a unit change in $\Delta\tau$, $\lambda^\star$ of Eq. 10, is broadly consistent with predictions by the $\tau$-$\omega$ model of Sec. 4.1.2. c) The additive biases are of the right direction (positive), but the unnormalized quantities $\mu^\star$ smaller than predicted by the model.

be small, usually on the order of 0.03 m$^3$ m$^{-3}$ (Fig. 5). These values are not directly comparable to standard RMSE estimates because our approach disentangles the quasi-random noise from a sensitivity that deviates from 1, temporally variable biases associated with $\Delta\tau$ and in-situ errors. Also, the quasi-random errors tend to increase with $\Delta\tau$ (Fig. 5), as predicted by the $\tau$-$\omega$ model (Fig. 3a). However, there is considerable variability in the magnitude across the study sties.

5   **4.2.2   Sensitivity analysis**

Our sensitivity analyses focus on the reference $\tau$ product. When the SMAP dual channel result is used as the reference $\tau$ product, the bias parameters change little for the vast majority of sites (Fig. 6, column 1). When the posterior uncertainties are taken into account, the $\lambda$ and $\mu$ values tend to overlap with those obtained using the SMOS $\tau$ product. This is despite potential disadvantages of the SMOS product (different resolution and footprint than SMAP, different retrieval model and use of NDVI-

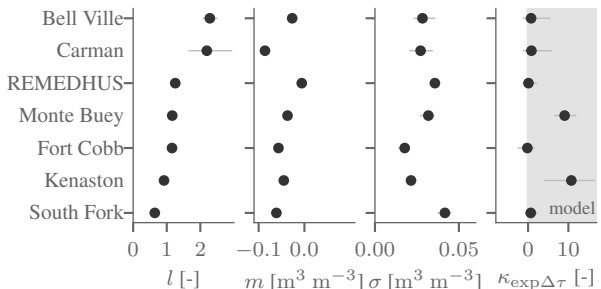

**Figure 5.** The time-average error properties $l$ (sensitivity), $m$ (additive bias) and $\sigma$ (noise standard deviation), and the $\kappa$ parameter (sensitivity of SMAP's noise level to $\Delta\tau$) as inferred for all network sites. Meaning of markers and lines as in Fig. 4.

based regularization in the retrieval (Al Bitar et al., 2017)) or of the SMAP DC product (e.g. no multi-angular information, measurement noise-induced error correlations with soil moisture estimate). Taken together, the similarity over the network sites indicates that the results are not particularly sensitive to whether the SMOS or the SMAP DC $\tau$ product serves as the reference for $\Delta\tau$.

By contrast, the estimates can change substantially when $\tau$ is derived from contemporaneous NDVI data, and predominantly they are smaller in magnitude (Fig. 6, column 2). If the problem with the use of the NDVI climatology in the retrieval were the use of a climatology alone, we would expect similar estimates. Conversely, we would expect the estimates to be smaller in magnitude if it was the link between NDVI and $\tau$ that led to an inaccurate vegetation correction, because the NDVI-based $\Delta\tau$ would provide little information on the biases. The smaller estimates that were actually observed may thus indicate that the use

of a climatology is not a dominant error source in the SMAP vegetation input data.

    The estimates of the time-variable biases are reasonably robust to other aspects of the model specification. The impact of replacing the MERRA2 with the GLDAS2 soil moisture or dropping it altogether is also small (Fig. 6, columns 3 and 4). By additionally including the smoothed SMOS $\tau$ as an explanatory variable, potential confounding on $\Delta\tau$ could be assessed: the estimates changed very little for all but one station (Fig. 6, column 5). When allowing the soil moisture parameters $A$ and $B$

to vary seasonally (Eq. 5), the parameter estimates do not change substantially for all sites but one (Fig. 6, column 6). The standard setup of the probabilistic model hence seems adequate for quantifying the SMAP error structure.

### 4.2.3   Sparse sites

Across the sparse sites within the contiguous US we commonly find pronounced time-variable biases (Fig. 7a-b). While the results over the sparse sites are deemed much less reliable than those over the network sites, they are similar in areas of overlap.

The largest changes in sensitivity of $\lambda \approx 0.3$ are found over croplands (e.g. Midwest, Central Valley in California). As predicted by the model, the $\lambda$ are predominantly positive but notable exceptions occur in the Mississippi Delta. Large and positive $\lambda$ are also common over pastures and grasslands. The additive bias parameters $\mu$ tend to show a similar spatial pattern, in that they are largest over crop- and grasslands (Fig. 7b).

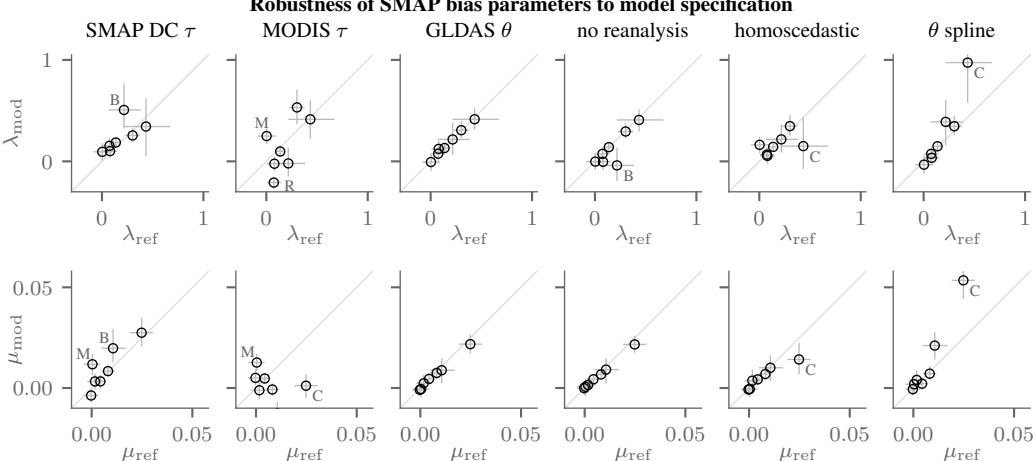

**Figure 6.** Robustness quantified by changes in the estimated $\lambda$ (top row, [-]) and $\mu$ (bottom row, [$m^3\ m^{-3}$]) parameters. Each panel compares the reference estimates on the horizontal axis (median: marker, lines: 10–90% posterior probability interval) to those obtained with the modified model on the vertical axis (cf. Sec. 4.1.2). The most prominent outliers are annotated – B: Bell Ville, C: Carman, M: Monte Buey, R: REMEDHUS.

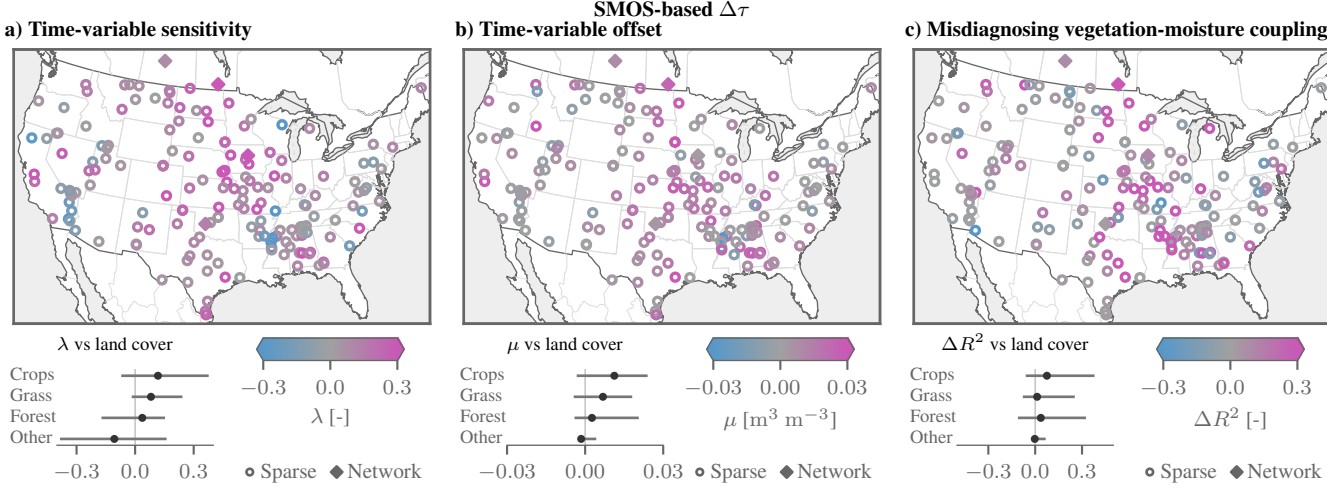

**Figure 7.** Time-variable biases and $\Delta R^2$ coupling metric across the contiguous US obtained using the SMOS $\tau$ product, shown for the network sites and the sparse sites (only those with >250 valid SMAP observations). a) The time-dependent sensitivity parameters $\lambda$ are predominantly positive, and the largest magnitudes are found over crop- and grasslands. b) The additive biases exhibit a similar spatial pattern. c) The degree of association $R^2$ between anomalies of vegetation $\tau$ and soil moisture is commonly higher for SMAP than for in-situ soil moisture ($\Delta R^2 > 0$).

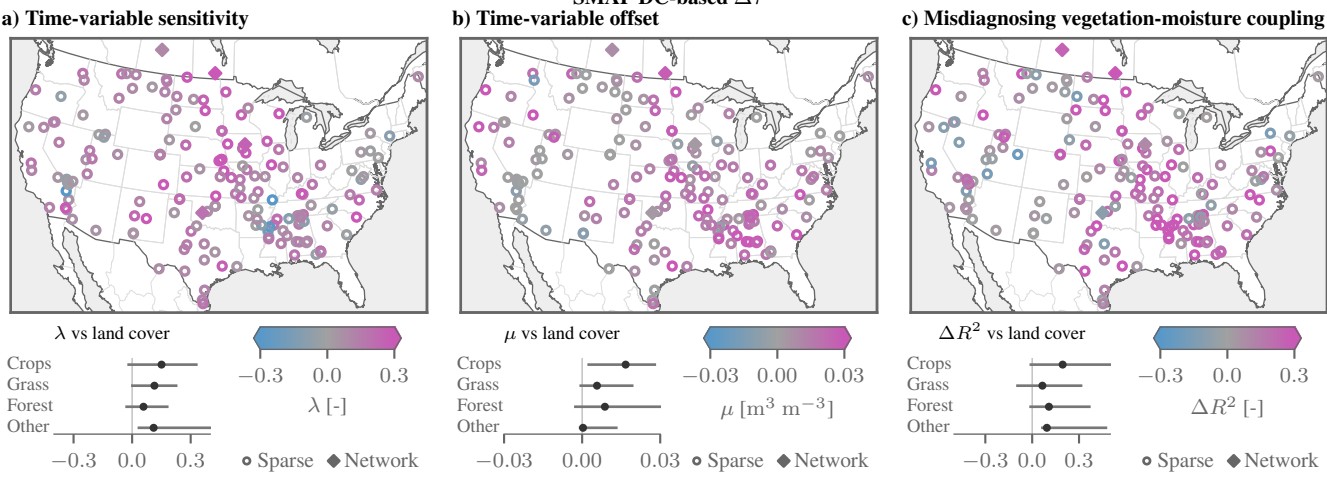

**Figure 8.** Time-variable biases and $\Delta R^2$ coupling metric across the contiguous US, obtained using the SMAP DC $\tau$ product. Cf. Fig. 7 for results based on the SMOS $\tau$ product.

Large biases over crop- and grasslands are also found when $\Delta \tau$ is computed from the SMAP DC product (Fig. 8a,b). The spatial patterns are very similar. Again, there is a close correspondence between the sparse and the network sites. Thus, irrespective of the $\Delta \tau$ product, the analysis of the sparse sites reveals sizeable biases over grasslands in addition to croplands.

To analyse the estimated noise level for all three products, we computed a normalized version $\sigma\, l^{-1}$, where the division by $l$ accounts for the different dynamic ranges of the three products by scaling the noise level with respect to the in-situ data (Fig. 9; SMOS-based $\Delta \tau$). SMAP achieves a median value of $0.045\,\mathrm{m^3\,m^{-3}}$, a higher value than that of the in-situ data or MERRA-2 ($0.029$ and $0.040\,\mathrm{m^3\,m^{-3}}$, respectively). For all three products, the corresponding values over the network sites are smaller by $\sim 50\%$. For the in-situ data, the larger noise level at the sparse sites is not surprising, owing to their limited representativeness. However, direct comparisons could be misleading. For instance, the larger noise level estimates (and greater spread of these estimates) may be partially accounted for by the small number of available networks and by the heterogeneous land cover and vegetation conditions across the sparse sites in the contiguous US.

### 4.2.4 Vegetation-soil moisture coupling

The observed coupling between vegetation and soil moisture anomalies is larger when using SMAP than when using in-situ soil moisture data (Fig. 7c). Positive values of $\Delta R^2$ are particularly pronounced over croplands (0.12 on average), with the spatial pattern largely conforming to that of the time-variable biases. Conversely, if the time-variable errors in the remotely sensed soil moisture were purely random, the degree of association would decrease, i.e. $\Delta R^2 < 0$. Even though spatial representative errors are likely large for the sparse networks, the $\Delta R^2$ values are comparable at proximal network sites. When the SMAP DC $\tau$ product is used instead, the results are similar in terms of both magnitudes and spatial patterns (Fig. 8c).

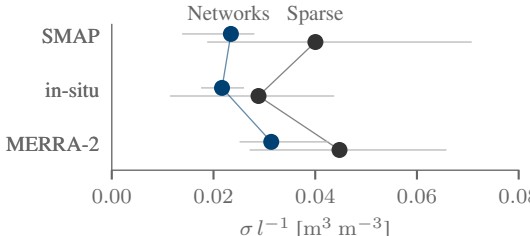

**Figure 9.** Estimated noise level normalized to the in-situ dynamic range. For both the network and the sparse in-situ sites, the distribution of the posterior median values of $\sigma l^{-1}$ is summarized by the median (marker) and the interquartile range (horizontal bar).

While the spatial patterns largely match those of the time-variable biases, the link between them is not clear and not necessarily uniform across all sites. The computation of anomalies largely removes seasonal offsets, which constitute a major fraction of the estimated additive biases. However, it does not remove higher-frequency variations or inter-annual differences, although the record is too short to reliably study those. Neither can it account for the changes in sensitivity, which are particu-
larly large over croplands. Finally, the in-situ soil moisture anomalies, predominantly derived from single probes, are subject to major uncertainties. All these factors likely contribute to the elevated associations between the $\tau$ and the SMAP soil moisture anomalies ($\Delta R^2 > 0$, but the precise impact of time-variable biases on our ability to diagnose such interactions remains an open question.

## 5  Discussion

### 5.1  Detected time-variable errors in the SMAP product

By applying Bayesian triple collocation to the SMAP soil moisture product, we detect time-variable biases. These time-variable biases track the misspecification of the vegetation optical depth $\Delta\tau$ during the soil moisture retrieval. They are both additive and multiplicative, i.e. not only the offset but also the sensitivity changes over time. Especially the changes in sensitivity can be large over croplands, as seasonal variations in $L$ on the order of 30% ($\lambda \approx 0.3$) deserve attention in future studies (Fig. 4,
7). The spatial patterns of the time-variable biases largely match those of the temporal autocorrelation estimated by Dong et al. (2018), which were also largest over croplands. These and our findings suggest that the NDVI-based vegetation correction in the SMAP retrieval introduces particularly large errors in agricultural regions.

A mechanistic interpretation of the inferred biases is complicated by a number of poorly understood factors. First, the time-variable biases are relative to the in-situ data. The results over the sparse sites should hence be interpreted with caution
due to representativeness error, even if they are similar to those at the dense high-quality network sites (Fig. 7). Also at the network sites, residual time-dependent biases in the in-situ data cannot be ruled out completely. Another major uncertainty are errors in the satellite-derived $\tau$ products, which are not accounted for in the estimation. One reason why these errors are

difficult to quantify is that in the context of soil moisture retrieval $\tau$ can be considered as essentially a model-internal effective quantity (Parrens et al., 2017). As such, an observation-based estimate of $\tau$ reflects not only the vegetation conditions but also inaccuracies of the tau-omega model itself, the way it is parameterized and other environmental conditions. An instance for the latter are roughness changes associated with harvest in croplands (Patton and Hornbuckle, 2013), which likely contribute

to the autumnal increase in SMOS $\tau$ in Fig. 4a. To a good degree of approximation, roughness changes will be captured by the effective $\tau$ that the SMOS or SMAP DC algorithms retrieve from the brightness temperatures (Parrens et al., 2017). Nevertheless, the estimates used in this study will still be affected by errors with respect to this effective quantity that can be random and systematic (e.g. due to the different incidence angles for SMOS and SMAP). The impact of such errors on the estimated biases is unknown, but analogies to simple regression models suggest that they can distort these estimates in either

direction (Frost and Thompson, 2002).

    While it is premature to attribute the inferred biases completely to an imperfect vegetation correction, there are two lines of reasoning that suggest that the inferred biases are not spurious. First, they are fairly consistent across croplands, and also between sites with sparse and dense in-situ networks (Fig. 7). Also, they tend to be large both in absolute terms (e.g. $\lambda > 0.1$) and compared to the posterior uncertainties. Further, they are also robust to the specification of the input $\tau$ product (SMAP

DC instead of SMOS) and to several model modifications (Sec. 4.2.2). However, these results are purely descriptive in that they only quantify associations, rather than establishing a causal link. A first step towards such a mechanistic interpretation is the comparison of the time-variable biases with predictions by the $\tau$-$\omega$ model. This second line of reasoning suggests that the estimated multiplicative biases $\lambda^\star$ are largely consistent with theoretical expectations (Fig. 4). However, this analysis is contingent on i) the tau-omega model being appropriate and correctly specified (e.g. known $\omega$), ii) there being no confounding

biases such as seasonal inundation, and iii) the sufficient accuracy of the input $\tau$ product. It is difficult to dispel these concerns, and indeed the deviations from the predictions (for $\mu^\star$) may indicate that unconsidered phenomena also contribute to the time-varying biases in addition to those resulting from the vegetation correction.

    One further caveat is that also time-average biases are present (additive bias: $m \neq 0$, sensitivity: $l \neq 1$). For instance, the SMAP retrievals at the South Fork site have too low a sensitivity ($l < 1$) and are too dry (negative $m$) on average (Fig. 4a).

Our analysis has focused on the time-dependent biases, partly because the time-independent biases are better known and more commonly compensated for in analyses such as data assimilation studies (Yilmaz and Crow, 2013; Kornelsen and Coulibaly, 2015; Colliander et al., 2017). Also, there are many potential sources for these time-invariant biases between the retrievals and the in-situ data, e.g. the calibration of the in-situ probes, the dielectric mixing model in the retrieval, or an offset in the mean input vegetation optical depth (i.e. $\mathrm{mean}(\Delta\tau) \neq 0$).

We conclude that our key finding is the presence of sizeable time-variable biases in the SMAP product. They are associated with, but likely not entirely caused by, deviations of the a priori $\tau$ used in the soil moisture retrieval from $\tau$ estimated from contemporaneous microwave data.

## 5.2 Implications for hydrological applications

The time-varying biases can have a negative impact in many applications. The changing sensitivity impedes the seasonal comparison of soil moisture dynamics, as the same SMAP-observed change corresponds to a wide range of actual soil moisture changes depending on the season (e.g. Fig. 4a). Variable sensitivities are particularly problematic for characterizing droughts, as extreme conditions may not be apparent in the SMAP data when the sensitivity happens to be small (as would happen for reduced vegetation water content). These issues also carry over to inter-annual comparisons. Inter-annual differences in the vegetation water content may, owing to the use of a climatology for $\tau$ in the retrieval, induce a spurious vegetation signal in the soil moisture retrievals.

The spurious vegetation signal in the soil moisture data may distort estimates of water-vegetation coupling. We find inflated values of $R^2$ between the SMOS vegetation optical depth and SMAP soil moisture, whereas purely random noise would decrease the $R^2$ (Fig. 7c). While the spatial patterns largely match those of the estimated biases, this does not imply a causal link between the two. However, the inflated $R^2$ values hint at potential pitfalls in using remotely sensed soil moisture to study global hydrology.

## 5.3 Estimating complex error structures

Soil moisture products can be subject to complex, time-variable errors, as revealed by our novel method for estimating such complex error structures from data. Other estimation procedures are conceivable, especially if high-quality in-situ data are available, and should be explored in the future. Our Bayesian triple collocation approach is widely applicable because it yields consistent estimates of error magnitudes and biases even when no error-free reference data set is available. It does, however, have to be assumed to be free of systematic error. The method is flexible, so that the error structure parameterization can be adapted to the problem at hand. We hope that this will enable the community to better characterize the uncertainties of remotely sensed soil moisture products. The knowledge of time-variable structural errors is key to improving the products, and it also helps to inform the application of these data sets in practice.

The presented approach could be applied to a wide range of variables besides soil moisture, such as wind speed, land surface fluxes and leaf area index. The issue of non-constant error sources, be they associated with environmental conditions or varying observational parameters, likely pertains to many such variables. By shedding light onto residual biases, our approach could in the future contribute to the development of improved retrieval approaches.

## 6 Conclusions

We developed a probabilistic approach for estimating complex error structures to study time-variable biases in the SMAP soil moisture product. We hypothesized that temporal changes in the error structure arise due to an inaccurate vegetation correction in the retrieval, so that the biases relative to the in-situ data track the misspecification in the vegetation optical depth $\Delta\tau$. Our conclusions are as follows

1. Sizeable temporal changes in the offset and the sensitivity were detected, and they were particularly large over croplands (e.g. change in sensitivity $\sim 30\%$).

2. While the estimated time-variable biases track the $\Delta\tau$ (with respect to contemporaneous SMOS or SMAP dual channel estimates), they are not necessarily entirely caused by an inappropriate vegetation correction. The attribution to this source is complicated by the potential presence of confounding variables or errors in the reference $\tau$. However, the time-variable multiplicative biases match the expected biases, as predicted by the $\tau$-$\omega$ model, in direction and magnitude, suggesting that the vegetation correction is indeed an issue. Conversely, the additive biases only match in direction.

3. The time-variable biases impede the seasonal comparison of remotely sensed soil moisture values. In particular, extreme conditions like droughts may not be apparent in the SMAP data when the sensitivity happens to be small.

4. The presented estimation approach is widely applicable because it yields consistent estimates of error magnitude and biases even when no error-free reference data set is available. Further, it is flexible in that a wide range of different kinds of error structures can be estimated purely from observations.

More generally, our findings illustrate the importance of recognizing time-variable biases in soil moisture products. These products are deemed central to a wide variety of tasks, including policy-relevant research on drought monitoring and the role of vegetation in weather, climate and the economy. The time-variable biases we detect can greatly reduce the extent to which the products can contribute to such tasks, unless they are accounted for. For instance, the distorted estimates of plant-water coupling we find for the SMAP product suggest that naive analyses of how water availability is coupled with vegetation growth and thus crop yields may give rise to misleading results. As similar phenology-dependent biases may also be present in other remotely sensed products, it is necessary to quantify and account for such seasonal and inter-annual biases when using the products in ecohydrology, agronomy and related applications.

Time-variable biases should thus be considered in future uncertainty analyses. Previous mission requirements have largely focused on the RMSE error metric, which, however, cannot distinguish between such systematic errors and white noise. Because neglecting that distinction can easily give rise to misleading interpretations, it is important that time-variable biases be quantified. The robust estimation approaches like that developed here can help to quantify and mitigate these biases, and thus to exploit the full potential of observational data sets.

# 7 Acknowledgements

The authors are grateful to Kaighin McColl, Dara Entekhabi and his group, and the SMAP team for comments and suggestions. They thank Alexandra Konings and Wade Crow for insightful reviews. S.Z. acknowledges support from the Swiss National Science Foundation (P2EZP2_168789). The support of the Canadian Space Agency, Environment and Climate Change Canada and the Natural Sciences and Engineering Research Council of Canada is acknowledge for support of the Kenaston network. A contribution to this work was made at the Jet Propulsion Laboratory, California Institute of Technology, under a contract with the National Aeronautics and Space Administration. The University of Salamanca team's involvement in this study was

supported by the Spanish Ministry of Economy and Competitiveness with the project PROMISES: ESP2015-67549-C3 and the European Regional Development Fund (ERDF). The remotely sensed and reanalysis data are available at the URLs provided in the references (registration generally required). The sparse in-situ data can be found at *ismn.geo.tuwien.ac.at*. The SMAP network in-situ data are available from *https://nsidc.org/data/nsidc-0712* until October 2016; the 2017 data are expected to be uploaded in early 2018; until then, they are available from the authors upon request. The Python code for Bayesian triple collocation has been made available at *https://github.com/szwieback/BayesianTripleCollocation*.

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
