# Peer review of "Estimating time-dependent vegetation biases in the SMAP soil moisture product"

_Hydrology and Earth System Sciences, 2018_

## Referee Comment (RC1) · A. Konings (Referee) · 24 Feb 2018

This paper presents a new, extended Bayesian methodology for estimating errors of remotely sensed soil moisture. The model is inspired by triple collocation approaches (and their assumed linear error model), and in some sense, extends triple collocation to allow time-varying multiplicative and additive errors. This new methodology is then applied to show that the sensitivity of the SMAP soil moisture product is influenced by its mis-specification of the vegetation optical depth, and that this could artificially inflate estimates of vegetation, soil moisture coupling. This paper could become an important contribution to the literature – the point about SMAP is quite informative given the broad use of this dataset. Furthermore, the new error characterization technique is an important advance and could (or should) become widely used. I applaud the authors

for the careful testing of the method through a simulation study and several sensitivity analyses However, as currently written, the paper is frequently lacking in sufficient detail of the methodology employed to derive its results, as I've outlined below. In particular, for each figure in the paper, what is shown in each figure and especially how it was described must be explicitly described in the text. This is not currently the case for a majority of figures. These, and a few other major concerns outlined below, need to be addressed before it can be published.

Major Comments:

A) Figure 1b lists the soil moisture as an output. If I understand correctly from the text, an explicit best guess 'true' soil moisture timeseries is never determined. This is probably the conservative thing to do – I am sure the uncertainty would be quite wide. Nevertheless, some explicit discussion/warning about the fact that this Bayesian approach is primarily for determining error statistics, and that accompanying posterior true soil moisture timeseries may not be useful (or if the authors disagree with me, some justification on that, as that would obviously be very intriguing!), is warranted.

B) Figure 2 is unclear. How is the bias defined? And how can the RMSE be greater than posterior in right-most column of Figure 2b if sigma simulation values (Table 1) are positive?

C) Even though the units are the same, it is a little confusing to have both the RMSE/bias and posterior on the same axes in Figure 2b, since the former represent a *difference*. I suggest splitting this into two rows. Than in the row where you show the posterior, it would also be useful to include the uncertainty of the posterior (through violin pots if necessary) and how it compares to the prior uncertainty. Is it actually much tighter, or has the mean just shifted? The bottom of page 7 mentions that "Fig. 2b shows that the posterior standard deviations are..." but I only see the posterior represented by a single point.

D) How is Figure 3a calculated? Is this assuming perfect retrieval? It must be influenced by the type of soil (influencing the dielectric mixing model) in some way...Also, are the different lines different average levels of true tau or something else? Please mention this also in the caption and clarify the text. What happened to the tau = 0.1 line in figure b? Did you decide to no longer use it? All of these things should be explained!

E) Fig 3b: The small clarification on the definition of L and M (which falls out of the model equations pretty easily) is negated by how long it takes to understand the figure because what it shows is barely described in the text. I suggest just removing this part of the figure.

F) Looking at Figure 4a, it is not clear visually that L is actually more closely related to delta tau than to tau itself. Can the authors check the statistics on this (preferably at all sites)? As evidenced by the sensitivity analyses the authors needed to do, estimating tau a priori is pretty difficult. If indeed L is a better match to tau directly than to delta tau, it would be easier for the understandability of the paper, and arguably more useful for future researchers' intuition about spatio-temporal variations in SMAP baseline soil moisture sensitivity.

G) More on Figure 4: The caption mentions "The magnitude of the dependence for a unit change in delta tau, lambda* is consistent with predictions by tau-omega". This is a strong-ish claim to casually throw into a caption. First of all, I'm guessing that the grey bar is some sort of model prediction from tau-omega? This needs to be explained in the caption though. It's particularly unclear since the color between the word 'model' is different than that of the grey bar. As mentioned elsewhere, the paper does not explain how it arrives at these model predictions. This has to be explained somewhere for it to be a paper that has any chance of being reproducible. Also, presumably it would not be hard to make these model predictions site-dependent (e.g. changing soil texture, estimated albedo, mean tau) – why are they constant with time? Lastly, it's unclear exactly what's going on in the right-hand column. Is it just the left hand column divided by the average delta tau at each site? If so, given that delta tau is probably as uncertain as the performance of the new methodology in this application and given

that the resulting model – estimate mismatch is actually not particularly encouraging, I suggest just leaving this out. Lastly, it would be useful if there was some discussion about what the sites mean. Are the trends in lambda and mu across sites consistent with e.g. vegetation density or canopy type characteristics?

H) Page 10, L27: I don't see why the re-analysis data error should depend significantly on delta tau at all. Why is this assumption made?

I) The baseline SMOS VOD product is known to have significant issues, because it relies heavily on an LAI-based prior (see discussion in Fernandez-Moran et al, Remote Sensing 2017). The SMOS-IC product has been developed specifically to get around this and early results are looking favorable. It is not yet publicly available to my knowledge, but the authors are quite willing to share. However, I am not sure SMOS VOD is the best 'true' VOD here – it will differ from the underlying ideal SMAP values due to differences in footprint, orbit, etc between the two satellites. Thus, I suggest using VOD from the dual-channel algorithm (either the O'Neill et al once currently used in the sensitivity analysis or I'd be happy to share our MT-DCA retrievals, which have somewhat less high-frequency noise and spatially variable albedo) instead of the SMOS VOD. The point in Figure 6 about the role of using optical data vs using a climatology for VOD would work just as well even without the first column in the figure.

J) The discussion section would benefit from some more discussion about the greater implications of this new methodology. For example, this technique might work particularly well for triple collocation of land surface fluxes of water and carbon, where it is easy to imagine significant seasonality in the error terms. Do the authors agree?

K) Similarly, can the authors discuss the implications of the normalization in Eq. 6 for the interpretation of the results?

Minor Comments:

L) Page 2, line 32: See also Momen et al, JGR-B 2017

[Figure]

M) Page 3, line 5: You haven't defined delta tau here

N) Figure 2: it would be helpful to explicitly explain somewhere why there are no RMSE values in the no mu, no lambda, no kappa case. It would also be easier to read the axes if there were more horizontal tick marks in each row, and if the tick labels were repeated between part a and part b.

O) Section 2.1.3: You assume quite specific priors. Would be helpful to show these distributions in the supplementary material to give the reader a sense of what they look like

P) Page 7: I suggest defining the RMSE error with equation or at least separate symbol for clarity. It's easy to miss this definition in the middle of the writing, but integral to following the rest of the discussion

Q) Page 7, line 29: How is this calculated?

R) Figure 2: Suggest splitting this into three columns: one with posterior vs. prior distribution (in violin plots if necessary), then third column with bias and RMSE.

S) Figure 2: Need to make it clearer that the 'no kappa' and no mu, lambda, kappa' simulations are cases where still have that in forward model. This is very difficult to pick out from text as is.

T) Page 11, line 5: note that this reference is broken

U) Page 16, line 1 : The authors might want to cite Crow et al, GRL 2015 here, which showed this point quite convincingly for soil moisture –latent heat coupling

V) I don't think the subscript p is ever defined. Is this an index for the number of explanatory variables?

References Cited:

Crow, W. T., Fangni, L., Hain, C., Anderson, M. C., Scott, R. L., Billesbach, D., &

Arkebauer, T. (2015). Robust Estimates of soil moisture and latent heat flux coupling strength obtained From Triple Collocation. Geophysical Research Letters, 8415–8423. https://doi.org/10.1002/2015GL065929

Fernandez-Moran, R., Al-Yaari, A., Mialon, A., Mahmoodi, A., Bitar, A. Al, Lannoy, G. De, . . . Wigneron, J.-P. (2017). SMOS-IC: An alternative SMOS soil moisture and vegetation optical depth product, (March), Remote Sensing, 1–26. https://doi.org/10.20944/preprints201703.0145.v1

Momen, M., Wood, J. D., Novick, K. A., Pangle, R., Pockman, W. T., Mcdowell, N. G., & Konings, A. G. (2017). Interacting Effects of Leaf Water Potential and Biomass on Vegetation Optical Depth. Journal of Geophysical Research: Biogeosciences. https://doi.org/10.1002/2017JG004145
* * *

---

## Referee Comment (RC2) · W.T. Crow (Referee) · 9 Mar 2018

This paper describes and applies a new analysis technique to identify time-dependent biases present in remotely sensed soil moisture products. This represents a very significant methodological advancement in the tools available to examine the error structure of these products (indeed any remotely sensed product). The authors offer a compelling motivation their approach (i.e., as we start to use remotely sensed soil moisture data products for coupling applications, it is important that we develop a more sophisticated understanding of their underlying errors). In my view, this paper represents a major step in that direction and has the potential to impact a great deal of on-going research plans (including my own).

[Figure]

Creative Commons BY license logo

However, as is often the case with highly novel manuscripts, there are some important questions regarding the presentation and interpretation of results that needed to be cleared up prior to publication.

Wade Crow

1) What happens if there is error correlation between the explanatory variable (w) and the products (y)? There are credible reasons to suspect that this arises between the SMOS "tau" product and the SMAP L3 soil moisture product - particularly in agricultural areas. Both products suffer from a common dependence of the zero-order tau-omega emission equation and the assumption of temporally constant surface roughness. These assumptions are particularly problematic over cropland agriculture and their violation could easily induce correlated errors into both products.

A related issue is that the interpretation of SMOS tau products is known to be complicated in agricultural areas (see e.g., https://lib.dr.iastate.edu/agron_pubs/115). In fact, the "reference" SMOS tau time series shown in Figure 4a demonstrates questionable features. First, corn crop canopies (responsible for ∼60% of the land cover in the South Fork water shed) typically demonstrate a biomass plateau between growth stages R2 and R6, which in Iowa which corresponds (roughly) to between August 1 and September 15 later. This expected "plateau" is actually somewhat more consistent with the SMAP "input tau" than the SMOS "reference tau" plotted in Figure 4a. Second, the rise in SMOS tau after October 1 is almost certainly a roughness artifact associated with post-harvest tillage and not a real vegetation opacity signal. So, there are credible reasons to suspect that (at least some) of the dynamics in the "delta tau" results actually reflect error in the SMOS tau "reference" (versus the SMAP tau input).

I'm probably overstating the problems with SMOS tau product here, but the broader question is how results are impacted by the presence of (potentially non-independent) errors in the explanatory variable? Is it possible that the diagnosed time dependent vegetation bias is due (in part) to the presence of error in in the SMOS tau product?

2) Section 2.1.1 – While the notation presented here which suggests that all three soil moisture products are subject to the same error model, I couldn't find any discussion of retrieved error parameters for the other two soil moisture products (i.e., in situ and MERRA). In addition, there seems to be a break in symmetry in that the selected explanatory variable is relevant for only one product (SMAP L3) and Figure 1 seems to indicate that no explanatory is applied to the in situ product. One of the appealing facets of triple collocation is the symmetry in its treatment of all three products. Does the break in symmetry applied here (via the selection of a single explanatory variable) preclude the objective cross comparison of error results across all three products? Discussion of error results for the other two products would also help establish credibility of the approach (e.g., were in time dependent biases found in the MERRA product and did that analysis reflect the known superiority of the core network relative to the other two products?).

3) Section 2.1.2 - The auto-regressive nature of a soil moisture time series signal is (arguably) its most defining characteristic. Therefore, the application of a transformed white noise process in (5) as a temporal soil moisture model is jarring. Some discussion regarding the sensitivity of results to the lack of serial correlation in (5) is needed. It is hard to imagine that the retrieval of time-dependent bias parameters is not impacted at least somewhat by the neglect of serial auto-correlation in the soil moisture model.

4) Section 2.2 - I understand that the Bayesian interference applied here is a fairly standard statistical procedure; however, I think it would help the (general earth science) reader if the authors provided more expository detail on exactly how the MC chain is implemented to solve the Bayesian problem. I'm a little unclear, for example, on how time is handled in the analysis (i.e. the analysis conducted sequentially or as a batch process across all time?).

On a related point, I'm also not quite clear on how effective the triple collocation analogy is. For example, the decision to use N=3 products seem almost arbitrary (e.g., later on the analysis, the MERRA product is dropped with apparently minimal consequence).

Presumably, larger N equates to tighter posterior distributions; however, this is never clarified.

5) Section 4.1.1 – I had to read this section a couple of time before I realized that the in situ observations were directly used as one of the three products in the Bayesian analysis (and not withheld as some type of independent verification). Presumably, the in situ observations correspond to the "y_o" product in described in Figure 1; however, I'm not sure if that link is ever explicitly made. More clarification on this point would be helpful.

6) Section 4.2.1 – Here I missed something fairly basic. What exactly is meant by the "model" referenced in the 3rd paragraph of the section and the vertical shading in parts b) and c) of Figure 4? Presumably, the authors are referring to the tau-omega model sensitivity results shown in Figure 3. However, this is never quite made clear. In addition, it isn't clear to me exactly how the (site-independent) "model" bias parameters are calculated. As a result, I'm missing some of the insight provided by Figures 4b and 4c. Is the take-away message that, despite not being given explicit access to the tau-omega model, the Bayesian model recovers the same bias parameter results predicted by the tau-omega model? I recommend that the authors spend a little more time outlining the context behind (and the interpretation of) Figure 4.

7) Section 4.2.2 - The authors provide a nice sensitivity analysis which describes the impact of using a different tau reference on results (in the first two columns of Figure 6). In theory, this should go a long way in addressing my first point; however, (as with the case in Figure 4 above) I did not take away as much from this figure as I had hoped. The lack of sensitivity in the time-variation bias parameters to the use of a second tau references is reassuring. However, I don't quite follow why the large changes observations when using a contemporary MODIS tau indicates a lack of sensitivity to the use of MODIS tau climatology in the SMAP L3 retrievals. The delta tau generated by the MODIS contemporary minus climatology differences leads to significantly non-zero lambda and mu estimates - just not the same estimates as the application of "delta tau"

[Figure]

results generated relative to SMOS tau. How exactly does this support the conclusion that inter-annual tau anomalies are not a significant source of error? Some additional discussion on this point would be very helpful. I also think a fuller sensitivity discussion of results in Figure 6 here would likely go a long way towards addressing concerns I raised in my first point.

8) Section 4.2.4 – The author's link the results in Figure 7c to the presence of time-dependent errors identified in Figure 7a and 7b. However, there is a major difference in that Figure 7c results reflect climatological anomalies (lacking any seasonality) while results in 7a and 7b reflect time-dependent biases which (almost certainly) have a fixed seasonal component (which, of course, would not be reflected in an anomaly). Therefore, a substantial(?) fraction of the time dependent biases reflected in Figures 7a and 7b have no impact on anomaly results in Figure 7c. Given this, I'm unclear exactly what the relevance of Figures 7a and 7b is for the interpretation of Figure 7c (although, admittedly there does appear to be some spatial consistency across the sub-figures).

9) I also have two general comments concerning Figure 7. I'll present them as "comments" to reflect that I'm inclined to give the authors some latitude with how they respond to them:

A) The authors discuss spatial representative issues; however, the impact of upscaling a single, point-scale observation to the SMAP footprint scale should not be underestimated. While the point is never explicitly made in Chan et al. [2016]; however, a comparison of TC-based results in (their) Figures 7 and 9 suggests that the correlation between a single-point ground observation and grid-scale truth is approximately equal to that between ASCAT soil moisture retrievals and the same grid-scale truth. Given that there is strong reason to suspect that SMAP soil moisture products are significantly more precise than ASCAT products, a priori, I'd expect single-point ground observations to be a noisier source of grid-scale soil moisture than SMAP L3 retrievals over a great deal of the United States. Combined with the fact that there is likely some

error cross-correlation between SMAP L3 products and SMOS tau products (especially over agricultural sites. . .see my point #1 above), it seems possible that results in Figure 7c can be explained without the need to invoke the presence of time-dependent vegetation biases in the SMAP L3.

Chen, F., Crow, W.T., Colliander, A., Cosh, M.H., Jackson, T.J., Bindlish, R., Reichle, R.H., Chan, S.K., Bosch, D.D., Starks, P.J. and Goodrich, D.C. Application of triple collocation in ground-based validation of Soil Moisture Active/Passive (SMAP) level 2 data products. IEEE Journal of Selected Topics in Applied Earth Observations and Remote Sensing. 99:1-14. 10.1109/JSTARS.2016.2569998. 2016.

B) Point-to-grid upscaling issues associated with ground-based soil moisture observations are particularly daunting for agricultural landscapes. Most of the time the actual site isn't even located in a cultivated field (instead that are typically shunted into non-cultivated areas at the edges of the field). As a result, these measurements have no hope of capturing (often significant) inter-annual soil moisture variability associated with changes in planting, canopy development and crop development. Given the soil moisture ground measurement expertise among the co-authors, I'll defer to their judgment on this issue - but it does seem relevant to the interpretation of Figure 7c.

---

## Author Comment (AC1) · 30 Apr 2018

This paper presents a new, extended Bayesian methodology for estimating errors of remotely sensed soil moisture. The model is inspired by triple collocation approaches (and their assumed linear error model), and in some sense, extends triple collocation to allow time-varying multiplicative and additive errors. This new methodology is then applied to show that the sensitivity of the SMAP soil moisture product is influenced by its mis-specification of the vegetation optical depth, and that this could artificially inflate estimates of vegetation, soil moisture coupling. This paper could become an important contribution to the literature – the point about SMAP is quite informative given the broad use of this dataset. Furthermore, the new error characterization technique is an important advance and could (or should) become widely used. I applaud the authors for the careful testing of the method through a simulation study and several sensitivity analyses However, as currently written, the paper is frequently lacking in sufficient detail of the methodology employed to derive its results, as I've outlined below. In particular, for each figure in the paper, what is shown in each figure and especially how it was described must be explicitly described in the text. This is not currently the case for a majority of figures. These, and a few other major concerns outlined below, need to be addressed before it can be published.

We are grateful to Alexandra Konings for the insightful review. We have added numerous clarifications, as we outline in our response.

Major Comments:
A) Figure 1b lists the soil moisture as an output. If I understand correctly from the text, an explicit best guess 'true' soil moisture timeseries is never determined. This is probably the conservative thing to do – I am sure the uncertainty would be quite wide. Nevertheless, some explicit discussion/warning about the fact that this Bayesian approach is primarily for determining error statistics, and that accompanying posterior true soil moisture timeseries may not be useful (or if the authors disagree with me, some justification on that, as that would obviously be very intriguing!), is warranted.

The algorithm estimates the posterior distribution of the soil moisture at each time step, as indicated in Fig. 1. What it does not yield is a single best guess, but rather a posterior distribution, although an estimate of the location (e.g. mean) could easily be derived from the posterior distribution. We now describe this in more detail in the section on MCMC sampling:

Each sample consists of draws from the posterior distribution, or actually an approximation thereof, of all the unobserved random variables (Output in Fig. 1b). They comprise the parameter random variables (e.g. the time-dependent biases) as well the soil moisture time series, i.e. one value of \theta for each SMAP observation.

For the future, we agree that the application of this technique to product merging (i.e. estimating soil moisture by combining several products) is an interesting avenue to explore, thus building on related triple collocation results (e.g. Yilmaz, M. T., W. T.Crow, M. C.Anderson, and C.Hain (2012), An objective methodology for merging satellite- and model-based soil moisture products, Water Resour. Res., 48, W11502, doi:10.1029/2011WR011682.)

B) Figure 2 is unclear. How is the bias defined? And how can the RMSE be greater than posterior in right-most column of Figure 2b if sigma simulation values (Table 1) are positive?

We have now defined the bias in Eq. 7, and similarly the RMSE is now defined in a separate equation (Eq. 6). The text has similarly been extended, and so has the caption.

The dot refers to the posterior standard deviation, as we now make clear in the legend. It is also mentioned in the caption and in the text.

> C) Even though the units are the same, it is a little confusing to have both the RMSE/bias and posterior on the same axes in Figure 2b, since the former represent a *difference*. I suggest splitting this into two rows. Than in the row where you show the posterior, it would also be useful to include the uncertainty of the posterior (through violin pots if necessary) and how it compares to the prior uncertainty. Is it actually much tighter, or has the mean just shifted? The bottom of page 7 mentions that "Fig. 2b shows that the posterior standard deviations are" but I only see the posterior represented by a single point.

As we state in our response to point B), we believe there has been a misunderstanding due to our insufficiently clear wording: the dot represents the posterior standard deviation. We believe this confusion arose due to the bad wording in the legend, which we have fixed. We now denote the posterior standard deviation by s_p throughout (text and figure). We contend that these quantities are directly comparable: for instance, asymptotically the posterior standard deviation of a parameter coincides with its RMSE (in a frequentist setting), provided certain regularity assumptions apply.

The posterior standard deviation is indeed considerably smaller than the prior standard deviation, i.e. the data tighten the distribution of a given parameter. For instance, the posterior standard deviation of mu shown in Fig. 2, is <0.01 m3m-3 and thus more than an order of magnitude smaller than the prior standard deviation of 0.3 m3m-3. We hope the new figure that shows the prior distributions will help readers to gauge this difference (see point O).

> D) How is Figure 3a calculated? Is this assuming perfect retrieval? It must be influenced by the type of soil (influencing the dielectric mixing model) in some way. Also, are the different lines different average levels of true tau or something else? Please mention this also in the caption and clarify the text. What happened to the tau = 0.1 line in figure b? Did you decide to no longer use it? All of these things should be explained!

We have amended the figure accordingly. We make clear that the two tau levels are the prescribed tau in the forward simulation, which is now described in much more detail:

> To compute the predicted biases in Fig. 3a), we assumed the \tau-\omega model applied and was correctly specified (temperature, dielectric mixing model [Dobson; silt loam], single-scattering albedo \omega = 0.05, etc.). For a given value of \tau_{\mathrm{true}}, we simulated the V-polarized brightness temperatures for dry and wet soil moisture conditions. These brightness temperatures were in turn the basis for estimating soil moisture by inverting the \tau-\omega model using the wrong \tau_{\mathrm{inv}} as a function of \Delta \tau. For both dry and wet soil moisture conditions, the deviation was an estimate of the retrieval bias: their mean was taken to be an estimate of the offset M, whereas their difference allowed us to estimate L. When plotted against \Delta \tau,

M and L are increase nearly linearly and only show a weak dependence on $t\tau_{\mathrm{true}}$. The slope of this relation is thus well but not perfectly defined. We refer to the slopes as $\mu^{\star}$ (for M) and $\lambda^{\star}$ (for L), respectively. To account for the spread due to the slight curvature and dependence on $\tau$, we estimated the likely range of values by computing the slopes from the differences in L or M between five equally spaced values of $\Delta \tau$ (between -0.1 and 0.1), repeated for equally many values of $\tau_{\mathrm{true}}$ (between 0.1 and 0.6). The range of these values was $\lambda^{\star}_{\mathrm{pred}}$ in [2.0, 3.8] and $\mu^{\star}_{\mathrm{pred}}$ in [0.33, 0.65] m3m-3. These ranges will later be compared to data-driven estimates, thus providing a first-order assessment of the agreement between predictions and observations, despite the neglect of other retrieval errors.

Further, we state explicitly that no tau value was assumed to produce figure b. We still have kept the same colour and linestyle across the two subfigures, as the caption should make it sufficiently clear that there is no direct link between the lines.

E) Fig 3b: The small clarification on the definition of L and M (which falls out of the model equations pretty easily) is negated by how long it takes to understand the figure because what it shows is barely described in the text. I suggest just removing this part of the figure.

The reason we included this figure in the first place is because the interpretation of L caused no small degree of puzzlement when we presented preliminary results of this study. As we suspect that some readers will skip Sec. 2 and 3, we have included this figure and we also recapitulate the meaning of the parameters in the text. While we have kept the figure, we have amended the caption in the hope that it will facilitate its interpretation. The relevant part reads

Explanation of the bias terms, illustrated for a time-changing sensitivity L(t) and offset M(t). A varying sensitivity changes the response of the SMAP retrieval to a unit change in the true soil moisture. When it is larger than one, the SMAP data have a larger dynamic range than the true soil moisture (illustrated by the slope > 1 in the inset). The time-average value of L is l, and the temporal standard deviation of L is given by $|\lambda|$ (length of arrow). A variable M induces non-constant offsets, and the magnitude of its temporal variability is given by $|\mu|$. M > 0 corresponds to a positive offset (shown in the inset).

F) Looking at Figure 4a, it is not clear visually that L is actually more closely related to delta tau than to tau itself. Can the authors check the statistics on this (preferably at all sites)? As evidenced by the sensitivity analyses the authors needed to do, estimating tau a priori is pretty difficult. If indeed L is a better match to tau directly than to delta tau, it would be easier for the understandability of the paper, and arguably more useful for future researchers' intuition about spatio-temporal variations in SMAP baseline soil moisture sensitivity.

We believe a potential confounding by tau is an important concern, and we have included an additional scenario in Fig. 7 to address it.

This new scenario uses two explanatory variables for L and M, namely Delta tau and tau itself. As we write in the methods: "To account for a potential confounding of \tau itself, which may also have an effect on the bias estimates, we included the smoothed SMOS \tau as second explanatory variable for $L$ and $M$, referred to as$\tau control." As we subsequently describe in the results, the estimates of the Delta tau lambda and mu change very little for all stations but one. This indicates that the standard inference results are not strongly influenced by confounding from this source. Also, the lambda parameter corresponding to tau is considerably smaller than that of Delta tau: medians of 0.00 and 0.16, respectively (25th/75th percentiles: -0.06/0.02 vs. 0.05/0.33). We hence do not believe that an additional dependence on tau, given delta tau, is a major issue here. However, we have completely revised the discussion section and now talk at length about confounding.

We have also computed estimates using only tau as explanatory variable, as suggested above, but we do not show them in the revised manuscript. The results for the tau parameters are potentially subject to confounding due to Delta tau (see above). For the South Fork site, the impression that there is a stronger relation to tau is borne out by the data to only a limited extent. The lambda parameter estimates turned out to be (10-90% posterior interval):

- standard model, i.e. only delta tau: Delta tau lambda: 0.25-0.36
- only tau: tau lambda: -0.02 - 0.11

Across all network sites, the Delta tau lambda are consistent in the sense that the posterior medians are all positive, whereas for the only tau configuration they are almost equally distributed between positive (4/7) and negative (3/7) values.

G) More on Figure 4: The caption mentions "The magnitude of the dependence for a unit change in delta tau, lambda* is consistent with predictions by tau-omega". This is a strong-ish claim to casually throw into a caption. First of all, I'm guessing that the grey bar is some sort of model prediction from tau-omega? This needs to be explained in the caption though. It's particularly unclear since the color between the word 'model' is different than that of the grey bar. As mentioned elsewhere, the paper does not explain how it arrives at these model predictions. This has to be explained somewhere for it to be a paper that has any chance of being reproducible. Also, presumably it would not be hard to make these model predictions site-dependent (e.g. changing soil texture, estimated albedo, mean tau) – why are they constant with time? Lastly, it's unclear exactly what's going on in the right-hand column. Is it just the left hand column divided by the average delta tau at each site? If so, given that delta tau is probably as uncertain as the performance of the new methodology in this application and given that the resulting model – estimate mismatch is actually not particularly encouraging, I suggest just leaving this out. Lastly, it would be useful if there was some discussion about what the sites mean. Are the trends in lambda and mu across sites consistent with e.g. vegetation density or canopy type characteristics?

To clarify these issues, we have made several changes to the text and figure.

We have already described the extended description of the model predictions. There, we outline how we arrive at the range of model predictions displayed in the figure as well as the limitations. The reason for using time-independent model estimates is that these estimates are based on time series, i.e. multiple time instances (with changing Delta tau) are required to estimate a time-independent parameter like mu/mustar.

We have changed the colour of the word 'model' and amended the caption:

The decent model-estimation match only pertains to lambda, i.e. subfigure b), and we have revised the caption to make this crystal clear. About lambda, we write that the magnitude are "broadly consistent with predictions by the \tau-\omega model of Sec. 4.'. Conversely, "the unnormalized quantities \mu^{\star} smaller than predicted by the model."

The right-hand column shows the comparison of the un-normalized quantities to the model predictions. To make it easier for the reader to understand this panel, we now show how the un-normalized estimates are computed in a separate equation (10), and we have greatly extended the discussion of the model predictions.

Finally, we briefly discuss the apparent dependence on potential controls (like land cover). We discuss spatial patterns and the relation to land cover at much greater length in the subsection on the sparse sites. In this subsection, we write:

> There is no clear apparent dependence of $\lambda^{\star}$ on location or land cover properties; for instance, Monte Buey and Bell Ville are within < 100 km of one another, and despite the similarity in planted crops the latter's $\lambda^{\star}$ is considerably larger.

> H) Page 10, L27: I don't see why the re-analysis data error should depend significantly on delta tau at all. Why is this assumption made?

We now discuss our rationale for including the Delta tau explanatory variable in the bias model of the re-analysis data.

> The inclusion of a $\Delta \tau$-dependent bias for the reanalysis product is not driven by physical reasoning, but for statistical reasons. By controlling for the same explanatory variables for both products, the impact of potential confounders - e.g. a seasonal bias that is correlated with $\Delta \tau$ - on the bias estimates of the remotely sensed product can be reduced. If this were not done, the model would try to partially adjust the time-variable bias term of the remote sensing product to minimize the systematic differences to the re-analysis product, thus distorting these bias estimates.

> I) The baseline SMOS VOD product is known to have significant issues, because it relies heavily on an LAI-based prior (see discussion in Fernandez-Moran et al, Remote Sensing

2017). The SMOS-IC product has been developed specifically to get around this and early results are looking favorable. It is not yet publicly available to my knowledge, but the authors are quite willing to share. However, I am not sure SMOS VOD is the best 'true' VOD here – it will differ from the underlying ideal SMAP values due to differences in footprint, orbit, etc between the two satellites. Thus, I suggest using VOD from the dual-channel algorithm (either the O'Neill et al once currently used in the sensitivity analysis or I'd be happy to share our MT-DCA retrievals, which have somewhat less high-frequency noise and spatially variable albedo) instead of the SMOS VOD. The point in Figure 6 about the role of using optical data vs using a climatology for VOD would work just as well even without the first column in the figure.

We share the reservations with respect to the tau products. To better address them, we have made a number of changes. First, we discuss the SMAP DC results obtained over the network sites in more detail. In particular, we mention some of the issues associated with either product. Second, we now also show the SMAP DC results over the contiguous US, i.e. over the sparse sites (Fig. ?). As with the network sites, the results are very similar. Third, in response to Wade Crow's remarks, we have included a separate discussion section where we discuss errors in the tau products and their impact on interpreting the results in a descriptive and a causal framework.

Note that we continue to use the SMOS L3 product, as we hope that we are able to paint a more complete picture by showing the results obtained with two different products. Unfortunately, the other products mentioned are currently not publically available. We hope that the techniques developed in the manuscript will in the future contribute to elucidating the error structure of novel products such as the MT-DCA soil moisture.

> J) The discussion section would benefit from some more discussion about the greater implications of this new methodology. For example, this technique might work particularly well for triple collocation of land surface fluxes of water and carbon, where it is easy to imagine significant seasonality in the error terms. Do the authors agree?

This is a good point. We have included a separate discussion section, where we dwell on the implications for error characterization more generally.

> Geophysical products in general are potentially also subject to time-variable errors, so that the presented approach could be applied to variables such as wind speed, land surface fluxes and leaf area index. The issue of non-constant error sources, be they associated with environmental conditions or varying observational parameters, likely pertains to many such variables. Extensions of our approach could in the future shed light on the error properties of a wide range of products, thus contributing to the development of improved retrieval approaches.

> K) Similarly, can the authors discuss the implications of the normalization in Eq. 6 for the interpretation of the results?

We do so by comparing the normalized results with absolute (unnormalized ones: the quantities with an asterisk). We detail the associated changes to this point in our reply to point G).

Minor Comments:

L) Page 2, line 32: See also Momen et al, JGR-B 2017

We have added a reference to this paper.

M) Page 3, line 5: You haven't defined delta tau here

At the beginning of the paragraph, we now write 'We hypothesize that seasonal changes in the error structure arise due to an inaccurate vegetation correction in the retrieval, so that the biases relative to the in-situ data track the misspecification in the vegetation optical depth \Delta \tau.' This is not a precise definition, but it should suffice for the introduction.

N) Figure 2: it would be helpful to explicitly explain somewhere why there are no RMSE values in the no mu, no lambda, no kappa case. It would also be easier to read the axes if there were more horizontal tick marks in each row, and if the tick labels were repeated between part a and part b.

We have amended the caption accordingly. We have also changed the ticks and labels as suggested. Note that we have slightly redesigned the figure in line with other suggestions.

O) Section 2.1.3: You assume quite specific priors. Would be helpful to show these distributions in the supplementary material to give the reader a sense of what they look like?

Good idea, we have added a new figure (supplement).

P) Page 7: I suggest defining the RMSE error with equation or at least separate symbol for clarity. It's easy to miss this definition in the middle of the writing, but integral to following the rest of the discussion

Done.

Q) Page 7, line 29: How is this calculated?

We now state explicitly the dynamic range on which these calculations were based: "The sensitivity coefficients \lambda are retrieved with comparable precision: the RMSE of 0.05 corresponds to a differential bias between dry and wet conditions of around 0.01 m3 m-3 (assuming a soil moisture dynamic range sim 0.25 m3m-3"

R) Figure 2: Suggest splitting this into three columns: one with posterior vs. prior distribution (in violin plots if necessary), then third column with bias and RMSE.

We believe this comment is to do with our bad wording in that we referred to the posterior standard deviation as the posterior (uncertainty), see points B) and C).

While we agree in principle that showing the entire posterior distribution is a good idea, we believe the well-behaved unimodal distributions, which we have exclusively encountered in our analyses, warrant the restriction to location and dispersion parameters to summarize those posterior distributions.

S) Figure 2: Need to make it clearer that the 'no kappa' and no mu, lambda, kappa'

simulations are cases where still have that in forward model. This is very difficult to pick out from text as is.

done

T) Page 11, line 5: note that this reference is broken

We have added the year

U) Page 16, line 1 : The authors might want to cite Crow et al, GRL 2015 here, which showed this point quite convincingly for soil moisture –latent heat coupling

We are grateful for this remark, as we were not aware of the paper.

V) I don't think the subscript p is ever defined. Is this an index for the number of explanatory variables?

We now also define it explicitly ("sensitivity to the pth explanatory variable")

---

## Author Comment (AC2) · 30 Apr 2018

This paper describes and applies a new analysis technique to identify time-dependent biases present in remotely sensed soil moisture products. This represents a very significant methodological advancement in the tools available to examine the error structure of these products (indeed any remotely sensed product). The authors offer a compelling motivation their approach (i.e., as we start to use remotely sensed soil moisture data products for coupling applications, it is important that we develop a more sophisticated understanding of their underlying errors). In my view, this paper represents a major step in that direction and has the potential to impact a great deal of on-going research plans (including my own). However, as is often the case with highly novel manuscripts, there are some important questions regarding the presentation and interpretation of results that needed to be cleared up prior to publication.
Wade Crow

We thank Wade Crow for his insightful review. We have made numerous modifications to the manuscript: apart from two new figures, we discuss several key results in considerably more detail.

1) What happens if there is error correlation between the explanatory variable (w) and the products (y)? There are credible reasons to suspect that this arises between the SMOS "tau" product and the SMAP L3 soil moisture product - particularly in agricultural areas. Both products suffer from a common dependence of the zero-order tau-omega emission equation and the assumption of temporally constant surface rough-ness. These assumptions are particularly problematic over cropland agriculture and their violation could easily induce correlated errors into both products. A related issue is that the interpretation of SMOS tau products is known to be complicated in agricultural areas (see e.g., https://lib.dr.iastate.edu/agron_pubs/115). In fact the "reference" SMOS tau time series shown in Figure 4a demonstrates questionable features. First, corn crop canopies (responsible for 60% of the land cover in the South Fork water shed) typically demonstrate a biomass plateau between growth stages R2 and R6, which in Iowa which corresponds (roughly) to between August 1 and September 15 later. This expected "plateau" is actually somewhat more consistent with the SMAP "input tau" than the SMOS "reference tau" plotted in Figure 4a. Second, the rise in SMOS tau after October 1 is almost certainly a roughness artifact associated with post-harvest tillage and not a real vegetation opacity signal. So, there are credible reasons to suspect that (at least some) of the dynamics in the "delta tau" results actually reflect error in the SMOS tau "reference" (versus the SMAP tau input).

I'm probably overstating the problems with SMOS tau product here, but the broader question is how results are impacted by the presence of (potentially non-independent) errors in the explanatory variable? Is it possible that the diagnosed time dependent vegetation bias is due (in part) to the presence of error in in the SMOS tau product?

The impact of errors in the tau product on the estimated bias parameters is an important concern that we now address in more detail throughout the manuscript. The main changes are extended discussions and a greater focus on an alternative tau product (sensitivity analysis based on the SMAP tau product) through extended discussions and a new figure. We are aware that we cannot resolve the issues mentioned, a fact we acknowledge openly, but we hope the extended discussions will provide a balanced picture.

There is now a separate discussion section that deals with the interpretation of the inferred biases. There, we posit that the role of errors when analysing the results is contingent on what general view one adopts. One of these views is purely descriptive, the other tries to establish a causal link. The purely descriptive view is easier to uphold because it is only concerned with associations rather than the mechanisms of these associations. As associations can be misleading if interpreted causally (errors in the tau product, confounding, etc.), we have clearly stressed the largely

descriptive nature of our analyses by employing phrases such as "errors associated with the vegetation correction" rather than induced or even caused.

It is the causal view that is more directly affected by errors in the input tau product. While we focus on a largely descriptive view, we do engage in analyses towards establishing a causal link, chiefly via the comparison to tau omega predictions. These comparisons rely on the assumptions of no confounding and no errors in the input tau product. We now mention these assumptions is explicitly (see below), and we discuss three important points in this context

> - the definition of the errors: in the context of soil moisture retrieval, we believe that it is mainly a model-internal effective parameter (that can partially account for e.g. changes in effective roughness or for an inappropriate choice of the effective scattering albedo). It is this effective parameter that should serve as reference in the computation of delta tau, rather than a purely vegetation-based proxy.

> - the nature of the errors, which have both systematic and random components

> - possible confounders

The distinction between a hypothetical true tau and an effective tau is, we believe, an important one to make, both for interpreting the estimated biases but also with the view of diagnosing of vegetation-water interactions that forms part of our motivation for studying time-varying biases. For single-channel retrievals, such a value typically exists: for a given soil moisture value (and forward model, single scattering albedo, etc.) it is the value that aligns the error-free brightness temperature with the true soil moisture. For dual-channel or multi-angular algorithms, such a value may not exist, in which case a retrieval of both tau and soil moisture would yield a wrong soil moisture estimate. However, it may be a good approximation, as hinted at by Parrens et al. 2017. They found that a joint retrieval of a single vegetation & roughness parameter, i.e. one effective tau parameter, yielded good soil moisture estimation results; this should work even better for a constant incidence angle (because in that case, the value of $N_r$ in their model is immaterial).

However, even if such an effective tau did exist, it would be dependent on the algorithm, incidence angle, etc. We stress this view in the discussion. We also highlight the limitations of this view in Sec. 5.2, where we discuss the diagnosing of vegetation-water interactions for which one tends to consider tau to be a vegetation proxy.

Owing to the complexities, we do not hazard a guess as to what influence deviations in the SMOS or SMAP DC tau from either a "true" or an "effective" tau have on the estimated biases. Based on analogies to regression modelling, we would expect causally biased estimates in the presence of random or systematic errors. We state this openly in the completely revised discussions. The two key paragraphs in the new discussion section read:

> A mechanistic interpretation of the inferred biases is complicated by a number of poorly understood factors. First, the time-variable biases are relative to the in-situ data. The results over the sparse sites should hence be interpreted with caution due to

representativeness error, even if they are similar to those at the dense high-quality network sites. Even at the network sites, residual time-dependent biases of the in-situ data cannot be ruled out completely. Another major uncertainty are errors in the satellite-derived contemporaneous tau products, which are not accounted for in the estimation. One important reason for why these errors are difficult to quantify is that in the context of soil moisture retrieval tau can be considered as essentially a model-internal effective quantity (Parrens17). As such, an observation-based estimate of tau reflects not only the vegetation conditions but also inaccuracies of the tau-omega model itself, the way it is parameterized and other environmental conditions. An instance for the latter are roughness changes associated with harvest in croplands (Patton13) which likely contribute to the autumnal increase in SMOS $\tau$ in Fig. 6a. To a good degree of approximation, roughness changes will be captured by the effective $\tau$ that the SMOS or SMAP DC algorithms retrieve from the brightness temperatures (Parrens17). Nevertheless, the estimates used in this study will still be affected by systematic and random errors with respect to this effective quantity. Systematic differences between the effective tau for the SMAP retrievals and that of the SMOS satellite are, for instance, due to different incidence angles and model parameters. The impact of such errors on the estimated biases is unknown, but analogies to simple regression models suggest that they can distort these estimates in either direction.

While it is premature to attribute the inferred biases completely to an imperfect vegetation correction, there are two lines of reasoning that suggest that the inferred biases are not spurious. First, they are fairly consistent across croplands, and also between sites with sparse and dense in-situ networks. Also, they tend to be large both in absolute terms (e.g. $\lambda > 0.1$) and compared to the posterior uncertainties. Further, they are also robust to the specification of the input tau product (SMAP DC instead of SMOS tau) and to several model modifications (Sec.5). However, these results are purely descriptive in that they only quantify associations, rather than establishing a causal link. A first step towards such a mechanistic interpretation is the comparison of the time-variable biases with predictions by the $\tau$-$\omega$ model. This second line of reasoning suggests that the magnitude of the multiplicative biases lambda* is largely consistent with theoretical expectations (Fig. 4a). However, this analysis is contingent on i) the tau-omega model being appropriate and correctly specified (e.g. known $\omega$), ii) there being no confounding biases such as seasonal inundation, and iii) the sufficient accuracy of the input tau product. It is difficult to dispel these concerns, and indeed the deviations from the predictions (for mu*) indicate that unconsidered phenomena also contribute to the time-varying biases in addition to those resulting from the vegetation correction.

2) Section 2.1.1 – While the notation presented here which suggests that all three soil moisture products are subject to the same error model, I couldn't find any discussion of retrieved error parameters for the other two soil moisture products (i.e., in situ and MERRA). In addition, there seems to be a break in symmetry in that the selected explanatory variable is relevant for only one product (SMAP L3) and Figure 1 seems to indicate that no explanatory is applied to the in situ product. One of the appealing facets of triple collocation is the symmetry in its treatment of all three products. Does the break in symmetry applied here (via the selection of a single explanatory variable) preclude the objective cross comparison of error results across all three products? Discussion of error results for the other two products would also help establish credibility of the approach (e.g., were in time dependent biases found in the MERRA product and did that analysis reflect the known superiority of the core network relative to the other two products?).

We now discuss some of the estimated parameters pertaining to the other products. First, we have added a new figure (Fig. 8) that shows the estimated noise level for all three products, or more precisely a normalized version that facilitates inter-product comparisons. In particular, the discussion addresses the spatial representativeness issues raised at several points in the referee report:

> To analyse the estimated noise level for all three products, we computed a normalized version sigma/l, where the division by l accounts for the different dynamic ranges of the three products by scaling the noise level with respect to the in-situ data (Fig. 8). SMAP achieves a median value of 0.045 m3m-3, a higher value than that of the in-situ data or MERRA-2 (0.029 and 0.040 m3m-3, respectively). For all three products, the corresponding values over the network sites are smaller by around 50%. For the in-situ data, the larger noise level at the sparse sites is not surprising, owing to their limited representativeness. However, direct comparisons could be misleading. For instance, the larger noise level estimates (and greater spread of these estimates) may be partially accounted for by the small number of available networks and by the heterogeneous land cover and vegetation conditions across the sparse sites in the contiguous US.

We also discuss MERRA mu/lambda parameters in a bit more detail, both in terms of the rationale and the results. With respect to the rationale, we have added that 'the inclusion of a delta tau dependent bias for the reanalysis product is not driven by physical reasoning, as the MODIS NDVI climatology that gives rise to a non-zero delta tau plays no role in the generation of the MERRA reanalysis product. However, there is a compelling statistical reason to include the same explanatory variables as in the remotely sensed product. By controlling for the same explanatory variables, the impact of potential confounders - e.g. a seasonal bias that is correlated with Delta tau - on the bias estimates of the remotely sensed product can be reduced. If this were not done, the model would try to partially adjust the time-variable bias term of the remote sensing product to minimize the systematic differences to the re-analysis product, thus distorting these bias estimates.' With respect to the results, we mention in the results that 'Reanalysis bias parameters were estimated as well, but they are considerably smaller in magnitude than those of the SMAP product.' More precisely, the mu parameter of the MERRA product is 0.000 on average (compared to 0.007 for the SMAP product), whereas for lambda they are 0.04 and 0.18 respectively. Further, their direction is highly heterogeneous, whereas the SMAP bias parameters are all of the same sign.

Finally, we do not share the view that the notation in Section 2.1.1. suggests that the error models are identical for all products. To include the possibility of different error models, we indexed the explanatory variables and the number of explanatory variables by the product. To better highlight this dependence, we now write "The explanatory variables can depend on the product n as well as on the parameter ($\mu$, $\lambda$). We also highlight that we use a reference product that is assumed unbiased.

> 3) Section 2.1.2 - The auto-regressive nature of a soil moisture time series signal is (arguably) its most defining characteristic. Therefore, the application of a transformed white noise process in (5)

as a temporal soil moisture model is jarring. Some discussion regarding the sensitivity of results to the lack of serial correlation in (5) is needed. It is hard to imagine that the retrieval of time-dependent bias parameters is not impacted at least somewhat by the neglect of serial auto-correlation in the soil moisture model.

We share those concerns, and we devised the simulation study to address some of them: in the simulations, the simulated soil moisture is auto-correlated, whereas the standard inference model implementation prescribes independent soil moisture values. However, we do agree that we did not discuss these aspects in sufficient detail. We now discuss the issue of time scales in the simulation section 3:

> The other crucial assumption in the model is the probability distribution for the soil moisture. Also here, the changes are typically small (up to 10% improvement in the RMSE, but a decrease in bias) when replacing the standard time-invariant model by a seasonally variable model. The improvement suggests that the model-internal soil moisture distribution can have an impact on the estimated bias parameters, in particular when the actual soil moisture is correlated with the explanatory variable, as it was in the simulated data. We would hence expect that for most applications it is the seasonal and sub-seasonal time scales that the soil moisture model should be able to capture. For comparison, autocorrelation on the inter-storm time scale that is not captured by our model but present in the simulated data did not seem to have a major impact (sufficient fidelity for the full model, Fig 2)

To expand on our discussion in the manuscript, we believe it is important to distinguish different time scales. The temporal structure of soil moisture time series is of course complex. To simplify it, we isolated two important time scales: long (i.e. seasonal) time scales, and short time scales

As outlined in our discussion, we believe that representing the seasonal time scales, on which also the biases vary in our application, is more important for accurate bias parameter inference than correctly representing the very short time scales. We do want to explore the shorter time scales in the future, though. It is definitely possible to represent such time scales in the model itself, but it requires a clever implementation (initially, we had tried to implement an AR-1 model for soil moisture, but it was impractical because the MCMC sampling was very slow, which is usually thought to indicate an issue with the way the model is set up. Internally, there are different ways to parameterize the same model, and they are not equivalent in terms of MCMC sampling efficiency).

> 4a) Section 2.2 - I understand that the Bayesian interference applied here is a fairly standard statistical procedure; however, I think it would help the (general earth science) reader if the authors provided more expository detail on exactly how the MC chain is implemented to solve the Bayesian problem. I'm a little unclear, for example, on how time is handled in the analysis (i.e. the analysis conducted sequentially or as a batch process across all time?).

To paint a clearer picture of the Bayesian approach, we have extended the description of the MCMC sampling.

… Here, we rely on Hamiltonian Monte Carlo as implemented using the adaptive No-U-Turn Sampler in pymc3. The No-U-Turn Sampler produces successive, dependent samples of the posterior distribution that are called a chain. Each sample consists of draws from the posterior distribution, or actually an approximation thereof, of all the unobserved random variables (Output in Fig1b). They comprise the parameter random variables (e.g. the time-dependent biases) as well the soil moisture time series, i.e. one value of \theta for each SMAP observation. For each location, we sample two independent chains with 2000 samples each, which standard quality controls (divergences, chain mixing) indicate is sufficient. Following common practice, the first 1000 samples are discarded

4b) On a related point, I'm also not quite clear on how effective the triple collocation analogy is. For example, the decision to use N=3 products seem almost arbitrary (e.g., later on the analysis, the MERRA product is dropped with apparently minimal consequence). Presumably, larger N equates to tighter posterior distributions; however, this is never clarified.

We agree that there are limitations to the analogy. Similar issues commonly arise when comparing classical inference approaches with Bayesian approaches. Classical approaches, including method-of-moments-type estimators that classical triple collocation can be thought an instance of, are plagued by problems of identifiability. In order for them to be applied successfully, the data must provide sufficient information to estimate all parameters at the same time, loosely speaking. In case of classical triple collocation, these are N=3 three error variances and often also N-1=two sets of bias parameters (often called additive and multiplicative bias), and they can be uniquely identified when N = 3 (but not when N = 2).

Conversely, for Bayesian approaches this issue does not arise in this form, owing to the prior information. A proper prior distribution (which is what we adopt in our approach) ensures valid posterior distributions even in the extreme case that no data are available, in which prior = posterior. A relevant publication in this context is Bayarri and Berger, The Interplay of Bayesian and Frequentist Analysis, Statistical Science, 2004.

In such data-poor situations the specific choice of prior plays a crucial role. It is to minimize the importance of the prior that we assumed 3 products, in analogy to triple collocation, as 3 products provide enough information even without any priors (classical case). For the scenario with only two products, we prescribed a much stronger prior on the in-situ data in an attempt to make up for the reduced information content.

We do realize that these qualitative arguments can only partially address the valid concern raised by the reviewer. However, we do not have theoretical results to bolster these views. The simulations studies indicate that N = 3 products is sufficient to estimate the parameters of interest (more precisely: to substantially narrow the posterior distribution compared to the prior). With respect to the appropriate choice of products (number, type, etc.), our manuscript leaves a lot of questions open.

To better address these concerns, we have provided an abridged summary of our rationale in Sec. 2:

> We focus on a setting inspired by triple collocation studies, i.e. we for the most part assume that N = 3 independent and noisy products are available Gruber 16. In regular triple collocation, three independent products provide sufficient information to estimate the random errors of all three products and bias parameters of two of the three products. In a Bayesian setting, the presence of prior information allows one to reduce the number of independent products, but the results will strongly depend on the prior distributions.

> 5) Section 4.1.1 – I had to read this section a couple of time before I realized that the in situ observations were directly used as one of the three products in the Bayesian analysis (and not withheld as some type of independent verification). Presumably, the in situ observations correspond to the "y_o" product in described in Figure 1; however, I'm not sure if that link is ever explicitly made. More clarification on this point would be helpful.

We have made two changes. First, we now list the three input products in the very first sentence of this section, and then describe them in more detail. Second, we now explicitly state that the in-situ data constitute the reference product y_0 (before, we had written that product n = 0 is the reference product).

> 6) Section 4.2.1 – Here I missed something fairly basic. What exactly is meant by the "model" referenced in the 3rd paragraph of the section and the vertical shading in parts b) and c) of Figure 4? Presumably, the authors are referring to the tau-omega model sensitivity results shown in Figure 3. However, this is never quite made clear. In addition, it isn't clear to me exactly how the (site-independent) "model" bias parameters are calculated. As a result, I'm missing some of the insight provided by Figures 4b and 4c. Is the take-away message that, despite not being given explicit access to the tau-omega model, the Bayesian model recovers the same bias parameter results predicted by the tau-omega model? I recommend that the authors spend a little more time outlining the context behind (and the interpretation of) Figure 4.

We have greatly extended the description and discussion of this aspect. The methods are now described in much greater detail. For M and L, we write

> To compute the predicted biases in Fig. 3a), we assumed the $\tau$-$\omega$ model applied and was correctly specified (temperature, dielectric mixing model [Dobson; silt loam], single-scattering albedo $\omega = 0.05$, etc.). For a given value of $\tau_{\mathrm{true}}$, we simulated the V-polarized brightness temperatures for dry and wet soil moisture conditions. These brightness temperatures were in turn the basis for estimating soil moisture by inverting the $\tau$-$\omega$ model using the wrong $\tau_{\mathrm{inv}}$ as a function of $\Delta \tau$. For both dry and wet soil moisture conditions, the deviation was an estimate of the retrieval bias: their mean was taken to be an estimate of the offset M, whereas their difference allowed us to estimate L. When plotted against $\Delta \tau$, M and L increase nearly linearly and only show a weak dependence on $t\tau_{\mathrm{true}}$. The slope of this relation is thus well but not perfectly defined. We refer to the slopes as $\mu^{\star}$ (for M) and $\lambda^{\star}$ (for L), respectively.

To account for the spread due to the slight curvature and dependence on \tau, we estimated the likely range of values by computing the slopes from the differences in L or M between five equally spaced values of \Delta \tau (between -0.1 and 0.1), repeated for equally many values of \tau_{\mathrm{true}} (between 0.1 and 0.6). The range of these values was \lambda^{\star}_{\mathrm{pred}} in [2.0, 3.8] and \mu^{\star}_{\mathrm{pred}} in [0.33, 0.65] m3m-3. These ranges will later be compared to data-driven estimates, thus providing a first-order assessment of the agreement between predictions and observations, despite the neglect of other retrieval errors.

We have also amended Fig. 3 (showing the star parameters explicitly). In the results, we have extended the description of the model-estimate comparison:

When converted into absolute quantities (\lambda^{\star}), the inferred dependence of L on \Delta \tau matches the model predictions reasonably well (Fig. \ref{fig:networksites}b). In other words, the data-derived, completely independent estimate is broadly consistent with the predicted impact of a \tau misspecification in the retrieval, despite limitations in the estimates (e.g. issues with the reference \tau) and the model predictions (e.g. assumed knowledge of the land surface temperature) of \lambda^{\star}. There is no clear apparent dependence of \lambda^{\star} on location or land cover properties; for instance, Monte Buey and Bell Ville are within < 100 km of one another, and despite the similarity in planted crops the latter's \lambda^{\star} is considerably larger.

We now also revisit this issue in the discussions; the associated changes to the appropriate section are described in our reply to point 1.

> 7) Section 4.2.2 - The authors provide a nice sensitivity analysis which describes the impact of using a different tau reference on results (in the first two columns of Figure 6). In theory, this should go a long way in addressing my first point; however, (as with the case in Figure 4 above) I did not take away as much from this figure as I had hoped. The lack of sensitivity in the time-variation bias parameters to the use of a second tau references is reassuring. However, I don't quite follow why the large changes observations when using a contemporary MODIS tau indicates a lack of sensitivity to the use of MODIS tau climatology in the SMAP L3 retrievals. The delta tau generated by the MODIS contemporary minus climatology differences leads to significantly non-zero lambda and mu estimates - just not the same estimates as the application of "delta tau" results generated relative to SMOS tau. How exactly does this support the conclusion that inter-annual tau anomalies are not a significant source of error? Some additional discussion on this point would be very helpful. I also think a fuller sensitivity discussion of results in Figure 6 here would likely go a long way towards addressing concerns I raised in my first point.

We have made two changes. First, we have added a new figure that also shows the results of the sparse sites obtained with the SMAP-based Delta tau. This figure also features in the second change, namely the extended descriptions. We also point out under what assumptions the smaller estimates obtained with the contemporaneous MODIS tau can be interpreted to indicate that the biases are not only due to outdated NDVI-derived data in the retrieval. In the results section, we write:

Our sensitivity analyses focus on the reference \tau product. When the SMAP dual channel result is used as the reference \tau product, the bias parameters change little for the vast majority of sites (Fig. 6). When the posterior uncertainties are taken into account, the \lambda and \mu values tend to overlap with those obtained using the SMOS \tau product, indicating that the results are not sensitive to the choice of microwave-derived reference \tau product. Also the spatial patterns across the sparse study sites are very similar (Fig. 8).

We have also extended the analysis of the MODIS-derived Delta tau.

By contrast, the estimates can change substantially when \tau is derived from contemporaneous NDVI data, and predominantly they are smaller in magnitude. If the problem with the use of the NDVI climatology in the retrieval were the use of a climatology alone, we would expect similar estimates. Conversely, we would expect the estimates to be smaller if it was the link between NDVI and \tau that led to an inaccurate vegetation correction. The smaller estimates that were actually observed may thus indicate that the use of a climatology is not a dominant error source in the SMAP vegetation input data.

As outlined in our reply to point 1), there is now a separate discussion section that deals with errors in tau in the context of interpreting the results.

8) Section 4.2.4 – The author's link the results in Figure 7c to the presence of time-dependent errors identified in Figure 7a and 7b. However, there is a major difference in that Figure 7c results reflect climatological anomalies (lacking any seasonality) while results in 7a and 7b reflect time-dependent biases which (almost certainly) have a fixed seasonal component (which, of course, would not be reflected in an anomaly). Therefore, a substantial(?) fraction of the time dependent biases reflected in Figures 7a and 7b have no impact on anomaly results in Figure 7c. Given this, I'm unclear exactly what the relevance of Figures 7a and 7b is for the interpretation of Figure 7c (although, admittedly there does appear to be some spatial consistency across the sub-figures).

We have extended the discussion of these results. Our main point in the discussions is not that there is a clear-cut link between the estimated biases and the R2 values, but rather that the spatial patterns suggest that there may be one that deserves attention in future studies.

In the results:

While the spatial patterns largely match those of the time-variable biases, the link between them is not clear and not necessarily uniform across all sites. The computation of anomalies largely removes seasonal offsets, which constitute a major fraction of the estimated additive biases. However, it does not remove higher-frequency variations or inter-annual differences, although the record is too short to reliably study those. Neither can it account for the changes in sensitivity, which are particularly large over croplands. Finally, the in-situ soil moisture anomalies, predominantly derived from single probes, are subject to major uncertainties. All these factors likely contribute to the elevated associations between the \tau and the SMAP soil moisture anomalies (Delta R2), but the

precise impact of time-variable biases on our ability to diagnose such interactions remains an open question.

In the discussion:

> The spurious vegetation signal in the soil moisture data may distort estimates of water-vegetation coupling. We find inflated values of $R^2$ between the SMOS vegetation optical depth and SMAP soil moisture, whereas purely random noise would decrease the $R^2$ (Fig. 7c). While the spatial patterns largely match those of the estimated biases, this does not imply a causal link between the two. However, the inflated $R^2$ values hint at potential pitfalls in using remotely sensed soil moisture to study global hydrology.

9) I also have two general comments concerning Figure 7. I'll present them as "comments" to reflect that I'm inclined to give the authors some latitude with how they respond to them: A) The authors discuss spatial representative issues; however, the impact of upscaling a single, point-scale observation to the SMAP footprint scale should not be underestimated. While the point is never explicitly made in Chan et al. [2016]; however, a comparison of TC-based results in (their) Figures 7 and 9 suggests that the correlation between a single-point ground observation and grid-scale truth is approximately equal to that between ASCAT soil moisture retrievals and the same grid-scale truth. Given that there is strong reason to suspect that SMAP soil moisture products are significantly more precise than ASCAT products, a priori, I'd expect single-point ground observations to be a noisier source of grid-scale soil moisture than SMAP L3 retrievals over a great deal of the United States. Combined with the fact that there is likely some error cross-correlation between SMAP L3 products and SMOS tau products (especially over agricultural sites see my point #1 above), it seems possible that results in Figure 7c can be explained without the need to invoke the presence of time-dependent vegetation biases in the SMAP L3.
Chen, F., Crow, W.T., Colliander, A., Cosh, M.H., Jackson, T.J., Bindlish, R., Reichle, R.H., Chan, S.K., Bosch, D.D., Starks, P.J. and Goodrich, D.C. Application of triple collocation in ground-based validation of Soil Moisture Active/Passive (SMAP) level 2 data products. IEEE Journal of Selected Topics in Applied Earth Observations and Remote Sensing. 99:1-14. 10.1109/JSTARS.2016.2569998. 2016.
B) Point-to-grid upscaling issues associated with ground-based soil moisture observations are particularly daunting for agricultural landscapes. Most of the time the actual site isn't even located in a cultivated field (instead that are typically shunted into non-cultivated areas at the edges of the field). As a result, these measurements have no hope of capturing (often significant) inter-annual soil moisture variability associated with changes in planting, canopy development and crop development. Given the soil moisture ground measurement expertise among the co-authors, I'll defer to their judgment on this issue - but it does seem relevant to the interpretation of Figure 7c.

We agree with the limitations of the sparse sites. This is also why we discuss the network sites in considerably more detail. Our motivation for including Figure 7c) is the similarity of the spatial patterns, which do suggest a connection. However, we make clear that the specifics of this link are currently unknown. In the future, we hope that our work on biases will inform the interpretation of correlation coefficients, regression models and similar statistics.

We hope the extended discussions, see in particular our reply to the previous point, clarify this stance.

---

## Author Response (AR2)

Dear Authors

Thank you for your patience during the review process. The paper can go forward for publication.

Both referees agree that you have now addressed their comments and that you have now produced a high quality research paper that has some important implications for modellers and practitioners in this field, and the users of their findings. In particular, your confirmation of the importance of recognising time-variable biases in soil moisture products and the shortcomings in commonly used methods in considering these has important implications in practice and for policy.

REgards

Graham

**We want to express our thanks to the editorial team and the two reviewers.**

Non-public comments to the Author:

The risk of misleading or distorted outcomes when applying the commonly used methods is critical. You have pointed this out in your comments outside of the paper. I think that this may be worth emphasising or stating more directly in the abstract. It's implied, but not stated directly.

When submitting the paper you stated that "Our findings have wide implications for policy-relevant research on drought monitoring and the role of vegetation in weather, climate and economy. Satellite-derived soil moisture products are deemed central to these tasks, but our work contends that time-variable biases can greatly reduce the extent to which satellite soil moisture products can contribute."

I think that a statement like this would not be out of place in the Conclusions and that you should consider adding one before final submission

- but I leave both suggestions up to you. The paper can proceed for publication, whatever you decide.

**We have modified the last two sentences of the abstract to make this point more explicitly. We have also rewritten the final paragraph of the conclusions to better highlight the implications for policy-relevant research.**

Please also note the minor corrections requested by one reviewer and consider these before final submission:

1) There is still no legend of what the line colors mean within figure 3a (only in the caption), which would be useful

We have labelled the two lines directly (rather than in a separate legend). The colours are also explained in the caption.

2) Page 20, line 10 seems to have a missing BibTeX reference

Fixed